# An Efficient Joint Learning Approach for Item Response Theory

**Tanish Agarwal** [* 1]  **Kaustubh Shivshankar Shejole** [* 2]  **Arpit Agarwal** [1 2]

## Abstract

Item response theory (IRT) is widely used in areas such as recommender systems, education, psychology, and other fields. A popular model for IRT is the Rasch model. Under this model, if a user with ability $\theta$ performs a task with difficulty $\beta$ then its label $X \sim \text{Bernoulli}(1/(1 + \exp(-(\theta - \beta))))$. Existing joint maximum likelihood estimation approaches for this problem do not perform well on small datasets and also lack theoretical guarantees. Recently, Nguyen & Zhang (2022) proposed a two step approach: (1) spectral method for estimation of task parameters, (2) likelihood optimization for estimation of user parameters. While this approach is theoretically sound, it is not computationally efficient. In this work, we propose an EM-based algorithm for joint estimation of item and user parameters by introducing Pólya-Gamma latent variables, which simplify the logistic log-likelihood. We show that our algorithm is both theoretically sound and consistently outperforms existing methods on synthetic and real-world datasets.[1]

## 1. Introduction

Item response theory (IRT) is a framework for modeling the responses to *test items* based on the *latent traits* of the individuals taking the test as well as the *latent traits* of the items themselves (Hambleton, 1991; Harvey & Hammer, 1999; Baker, 2001). IRT has traditionally been applied to educational testing (Hambleton, 1991), psychological

assessment (Harvey & Hammer, 1999) and health outcomes measurement (Cappelleri et al., 2014), however, it has also recently been applied to crowdsourcing (Whitehill et al., 2009), recommendation systems (Nguyen & Zhang, 2022), financial analysis (Schellhorn & Sharma, 2013), and more recently to the evaluation and benchmarking of language models (Lalor et al., 2016; Maia Polo et al., 2024; Hofmann et al., 2025).

Various IRT models have been proposed, differing in the number of parameters used to characterize items and individuals. The most popular model is the Rasch model (Hambleton, 1991; Schellhorn & Sharma, 2013; Nguyen & Zhang, 2022). Under this model, each user (or examinee) is characterized by a single parameter $\theta$ representing their ability, and each item (or task) is characterized by a single parameter $\beta$ representing its difficulty, and the probability that the user answers the item correctly is given by the logistic function:

$$\mathbb{P}(X = 1|\theta, \beta) = \frac{1}{1 + \exp(-(\theta - \beta))}. \tag{1}$$

Given responses from multiple users to multiple items, the goal is to jointly estimate the user and item parameters under the Rasch model.

Several methods have been proposed for estimation under the Rasch model. The joint maximum likelihood estimation (JMLE) (Haberman, 2004; Fischer, 1981) method jointly optimizes for both user and item parameters but it is known to produce inconsistent estimates (Ghosh, 1995). There have also been other methods such as marginal maximum likelihood estimate (MMLE) (Bock & Aitkin, 1981) and conditional maximum likelihood estimate (CMLE) (Haberman, 1977; Hambleton, 1991) which optimize the likelihood over the item parameters after integrating out user parameters or conditioning on sufficient statistics respectively. While these methods are better than JMLE, they are either prone to prior misspecification or are computationally expensive. Furthermore, since these approaches primarily estimate item parameters, one needs to do the estimation of user parameters in a separate step afterwards. More recently, (Nguyen & Zhang, 2022) proposed a method where they first estimate item parameters using a spectral approach and then estimate user parameters using likelihood-based estimates. While Nguyen & Zhang (2022) achieves theoretical bounds on

---

[*]Equal contribution  [1]Centre for Machine Intelligence and Data Science, IIT Bombay, India [2]Department of Computer Science and Engineering, IIT Bombay, India. Correspondence to: Tanish Agarwal <tanishagarwal2910@gmail.com, agarwaltanish@cse.iitb.ac.in>, Kaustubh Shivshankar Shejole <kaustubhshejole@cse.iitb.ac.in>.

*Proceedings of the $43^{rd}$ International Conference on Machine Learning*, Seoul, South Korea. PMLR 306, 2026. Copyright 2026 by the author(s).

[1]The implementation code is publicly available at https://github.com/agarwaltanish10/pgem-rasch-model

error, their error bound relies on assumptions that demand denser regimes i.e., many user-item labelings. A detailed discussion of related work is provided in Appendix A.

In this work, we aim to address the problem of jointly estimating both user and item parameters under the Rasch model in a computationally efficient manner that also performs well in sparser regimes. We introduce a novel technique based on Pólya-Gamma augmentation (Polson et al., 2013) where we introduce additional latent variables which transform the intractable log-likelihood associated with the Rasch model into a tractable Gaussian form. This results in an EM-algorithm where the E-step takes an expectation over the Pólya-Gamma latent variables, while the M-step optimizes a quadratic objective over the user and item parameters. The E-step can be evaluated in closed form using known expressions for the moments of Pólya-Gamma random variables (Polson et al., 2013), and the M-step reduces to solving a system of linear equations, which makes the algorithm computationally efficient.

Our theoretical analysis on the convergence of the proposed EM-algorithm shows that this algorithm achieves linear convergence to the MAP estimator. To our knowledge, this is the first theoretical guarantee for local convergence of the EM-algorithm for joint estimation under the Rasch model. Furthermore, we provide a finite-sample guarantee showing that the iterates converge linearly up to a tight statistical error. The experimental evaluations on synthetic and real-world datasets show that our algorithm is not only computationally efficient but also outperforms previous approaches. Our experiments demonstrate that compared to our method, previous methods perform poorly in sparser regimes, i.e., when there are only few valid user-task assignments or user-item labelings.

**Organization.** In Section 2, we introduce the problem of jointly learning user and item parameters under the Rasch model. Section 3 explains Pólya–Gamma augmentation for transforming the associated log-likelihood. In Section 4, we describe our Expectation–Maximization (EM) algorithm. Section 5 contains the theoretical analysis of our algorithm, and Section 6 reports experimental results on synthetic and real-world datasets.

## 2. Problem Formulation

**Notation.** For a positive integer $k$, we use $[k]$ to denote the set $\{1, 2, \ldots, k\}$. Vectors are denoted in bold, and $\mathbf{1}_d$ denotes the all-ones vector in $\mathbb{R}^d$.

We study a setting with $n$ users and $m$ items, where users provide *binary* ratings to items. Each user $l \in [n]$ is associated with a latent ability parameter $\theta_l^* \in \mathbb{R}$, and each item/task $i \in [m]$ is associated with a latent difficulty pa-

rameter $\beta_i^* \in \mathbb{R}$. We let $\phi = (\{\theta_l\}_{l=1}^n, \{\beta_i\}_{i=1}^m)$ denote the concatenated parameter vector. In IRT, the probability of a favorable response depends on both parameters. In this work, we adopt the celebrated Rasch model which specifies that, conditional on the latent parameters, the response of user $l$ to item $i$, $X_{li}$ independently follows a Bernoulli distribution with success probability

$$\Pr(X_{li} = 1) = \frac{1}{1 + \exp(-(\theta_l^* - \beta_i^*))}. \qquad (2)$$

The response data is represented by a matrix $X$, which may be only partially observed, as each user may respond to only a subset of the available items. Accordingly, $X \in \{0, 1, *\}^{m \times n}$, where $*$ denotes a missing entry. We represent the observation pattern via an indicator matrix, $M \in \{0, 1\}^{m \times n}$, where $M_{li} = 1$ if the response $X_{li}$ is observed, and $M_{li} = 0$ otherwise (that is, $X_{li} = *$). Each entry $X_{li}$ is observed independently with probability $p$, i.e., $\Pr(M_{li} = 1) = p$.

Given the (partially) observed binary response matrix $X$ generated according to the Rasch model, our goal is to efficiently recover both user abilities $\boldsymbol{\theta}^*$ and item difficulties $\boldsymbol{\beta}^*$. Note that this model is invariant under additive scaling of both parameters: for any $\alpha \in \mathbb{R}$, the vectors $\boldsymbol{\phi}^*$ and $\boldsymbol{\phi}^* + \alpha \mathbf{1}_{n+m}$ define the same likelihood. As a result, the model is not identifiable without additional constraints, and we impose the centering constraint $\sum_i \beta_i = 0$ to resolve this ambiguity.

## 3. Pólya-Gamma Augmentation

In this section, we describe the primary technique that we use in developing our EM algorithm. Following Equation (2), the complete-data likelihood can be written as

$$\mathcal{L}_{\text{complete}}(\boldsymbol{\phi}; X) = \prod_{(l,i)} \Big[ \big\{ p \cdot \sigma\big((-1)^{1+X_{li}}(\theta_l - \beta_i)\big) \big\}^{M_{li}}$$
$$\times (1-p)^{1-M_{li}} \Big].$$

Given that the likelihood contribution of missing entries depends only on $p$ and not on $\phi$, we restrict attention to the likelihood of the observed responses. Conditioning on $M$, this can be written as

$$\mathcal{L}_{\text{obs}}(\boldsymbol{\phi}; X) \propto \prod_{(l,i) \in \mathcal{O}} \frac{1}{1 + \exp\left\{(-1)^{X_{li}}(\theta_l - \beta_i)\right\}}, \quad (3)$$

where $\mathcal{O} = \{(l, i) \mid M_{li} = 1\}$ is the index set of observed entries.

However, direct maximization of the likelihood complicates theoretical analysis. To obtain a tractable optimization procedure, we leverage a data augmentation strategy introduced

by Polson et al. (2013), that is based on the *Pólya-Gamma* (PG) distribution.

Specifically, for a random variable $\omega$ that follows a PG$(b, 0)$ distribution with parameters $b > 0$ and $c = 0$, and for any real $\psi$, the following identity holds:

$$\frac{(e^\psi)^a}{(1 + e^\psi)^b} = \frac{e^{\kappa\psi}}{2^b} \int_0^\infty \exp\left(-\frac{\omega\psi^2}{2}\right) p(\omega \mid b, 0)\, d\omega, \quad (4)$$

where $\kappa = a - b/2$, and $p(\cdot)$ is the density of $\omega$. Further, the conditional density of $\omega$ given $\psi$ is again Pólya–Gamma with parameters $b$ and $\psi$, that is,

$$(\omega \mid \psi) \sim \text{PG}(b, \psi), \quad (5)$$

and all finite moments of a random variable $\omega' \sim \text{PG}(b, c)$ admit closed form expressions as

$$\mathbb{E}[\omega'] = \frac{b}{2c} \tanh\left(\frac{c}{2}\right). \quad (6)$$

Setting $a = 0$, $b = 1$, and $\psi = \eta_{li} := \theta_l - \beta_i$, the integral identity (4) allows us to rewrite the likelihood contribution of the $(l, i)$-th response as

$$p(X_{li} \mid \phi) = \frac{1}{1 + \exp\{(-1)^{X_{li}}(\theta_l - \beta_i)\}}$$
$$= \frac{1}{2} \exp\left(-\frac{(-1)^{X_{li}}\eta_{li}}{2}\right)$$
$$\times \int_0^\infty \exp\left(-\frac{(-1)^{2X_{li}}\omega_{li}\eta_{li}^2}{2}\right) p(\omega_{li})\, d\omega_{li}$$
$$= \frac{1}{2} \int_0^\infty \exp\left(-\frac{(-1)^{X_{li}}\eta_{li}}{2} - \frac{\omega_{li}\eta_{li}^2}{2}\right) p(\omega_{li})\, d\omega_{li},$$

where $\omega_{li} \sim \text{PG}(1, 0)$ is a latent Pólya-Gamma variable introduced for each user-item pair $(l, i)$. This induces the augmented joint density

$$p(X_{li}, \omega_{li} \mid \phi) \propto \exp\left(-\frac{(-1)^{X_{li}}\eta_{li}}{2} - \frac{\omega_{li}\eta_{li}^2}{2}\right) p(\omega_{li}), \quad (7)$$

which takes an exponential form. This transformation from a logistic to an exponential form enables the derivation of a simple and efficient EM algorithm, described in the following section.

## 4. EM Algorithm

The basis of our algorithm is as follows: using Equation (7), and given all $\omega_{li}$, the log-likelihood objective over $\phi$ simplifies to the following form (further details in Appendix C):

$$\ell(\phi) = -\frac{1}{|\mathcal{O}|}\sum_{(l,i)\in\mathcal{O}}\left[\frac{(-1)^{X_{li}}}{2}(\theta_l - \beta_i) + \frac{1}{2}\omega_{li}(\theta_l - \beta_i)^2\right]$$

This formulation naturally leads to an EM procedure where we first evaluate the expectation of the log-likelihood with respect to the latent variables conditioned on the current parameter estimates (E-step), and then maximize this expected log-likelihood to update the parameters (M-step). Observe that this expression is linear in $\omega_{li}$ and quadratic in $\phi$, giving the algorithm a simple structure. Algorithm 1 summarizes the resulting method, with the individual steps detailed below.

### 4.1. E-step

At iteration $t$, given the current estimate $\phi^{(t)}$, set $\eta_{li}^{(t)} := \theta_l^{(t)} - \beta_i^{(t)}$. We compute the expectation $\mathbb{E}[\, l(\phi) \mid \phi^{(t)}]$. Following identity (5), the variables $\omega_{li} \mid \phi^{(t)}$ follow the distribution PG$(1, \eta_{li}^{(t)})$, due to which we have

$$\mathbb{E}\left[\omega_{li} \mid \phi^{(t)}\right] = \mathbb{E}\left[\omega_{li} \mid \eta_{li}^{(t)}\right],$$

and the $Q$ function becomes

$$Q\left(\phi \mid \phi^{(t)}\right) = -\frac{1}{|\mathcal{O}|}\sum_{(l,i)\in\mathcal{O}}\left[\frac{(-1)^{X_{li}}}{2}(\theta_l - \beta_i)\right.$$
$$\left. + \frac{1}{2}\kappa_{li}^{(t)}(\theta_l - \beta_i)^2\right],$$

where, using identity (6), $\kappa_{li}^{(t)} = \mathbb{E}\left[\omega_{li} \mid \eta_{li}^{(t)}\right]$ is computed as

$$\kappa_{li}^{(t)} = \begin{cases} \dfrac{1}{2\,\eta_{li}^{(t)}} \tanh\left(\dfrac{\eta_{li}^{(t)}}{2}\right), & \eta_{li}^{(t)} \neq 0 \\ \dfrac{1}{4}, & \eta_{li}^{(t)} = 0 \end{cases}$$

### 4.2. M-step

In the Rasch model, the ML estimate may fail to exist when all observed responses associated with a user or item are identical, causing the likelihood to become unbounded and the corresponding parameter to diverge. Such situations can arise in small and/or sparse datasets.

To ensure well-posed estimation, we adopt a maximum a posteriori (MAP) formulation in our method. Specifically, we add an isotropic Gaussian prior on the parameter vector, $\phi \sim \mathcal{N}(0, \sigma^2 I)$, which induces an $\ell_2$ regularization term in the maximization objective. Writing $\nu := \frac{1}{\sigma^2}$, the M-step corresponds to maximizing a *regularized $Q$* function:

$$\phi^{(t+1)} \leftarrow \arg\max_\phi Q(\phi \mid \phi^{(t)}) - \frac{\nu}{2}\|\phi\|_2^2.$$

We denote the resulting regularized objective by $\widetilde{Q}$. The parameter updates are obtained by setting

**Algorithm 1** PGEM

**Input:** Binary response data matrix $X$ for $m$ items and $n$ users.

**Output:** Parameter estimates $\hat{\phi} = (\hat{\theta}, \hat{\beta})$

1: Initialize $\phi^{(0)}$
2: $t \leftarrow 0$
3: **repeat**
4:     **E-step:**
5:     $Q(\phi \mid \phi^{(t)}) \leftarrow \mathbb{E}_{\omega \mid X, \phi^{(t)}}[\ell(\phi)]$
6:     **M-step (Gauss-Seidel):**
7:     Initialize $(\theta^{(0)}, \beta^{(0)}) \leftarrow \phi^{(t)}$, $k \leftarrow 0$
8:     **repeat**
9:         $\beta^{(k+1)} \leftarrow \arg\max_{\beta} \widetilde{Q}(\beta, \theta^{(k)} \mid \phi^{(t)})$
10:        $\beta^{(k+1)} \leftarrow \beta^{(k+1)} - \frac{\mathbf{1}^T \beta^{(k+1)}}{m}$
11:        $\theta^{(k+1)} \leftarrow \arg\max_{\theta} \widetilde{Q}(\beta^{(k+1)}, \theta \mid \phi^{(t)})$
12:        $k \leftarrow k + 1$
13:     **until** convergence
14:     $\phi^{(t+1)} \leftarrow (\beta^{(k)}, \theta^{(k)})$
15:     $t \leftarrow t + 1$
16: **until** convergence
17: **return** $\phi^{(t)}$

---

$\nabla_{\phi} \widetilde{Q}\left(\phi \mid \phi^{(t)}\right) = 0$, which gives a relation that the optimal $\beta_i^{(t+1)}$ and $\theta_l^{(t+1)}$ must satisfy

$$\beta_i^{(t+1)} = \frac{\frac{1}{|\mathcal{O}|} \sum_{l \in T_i} \left( \kappa_{li}^{(t)} \theta_l^{(t+1)} + \frac{(-1)^{X_{li}}}{2} \right)}{\nu + \frac{1}{|\mathcal{O}|} \sum_{l \in T_i} \kappa_{li}^{(t)}}$$

$$\theta_l^{(t+1)} = \frac{\frac{1}{|\mathcal{O}|} \sum_{i \in S_l} \left( \kappa_{li}^{(t)} \beta_i^{(t+1)} - \frac{(-1)^{X_{li}}}{2} \right)}{\nu + \frac{1}{|\mathcal{O}|} \sum_{i \in S_l} \kappa_{li}^{(t)}}$$

Here, $S_l$ and $T_i$ are index sets defined as $S_l = \{i \mid (l, i) \in \mathcal{O}\}$ and $T_i = \{l \mid (l, i) \in \mathcal{O}\}$. Observe that the update for $\theta_l^{(t+1)}$ depends on $\beta_i^{(t+1)}$, and vice versa. Consequently, the update equations jointly define a linear system in $\phi$, which can be written as $H\phi = r$, where the definitions of $H$ and $r$ are provided in Appendix D. This system admits a unique solution, which corresponds exactly to the M-step update $\phi^{(t+1)}$. We compute this solution using the Gauss-Seidel (GS) method. Since the coefficient matrix $H$ is positive definite, convergence of the GS method, and thus of the M-step, is guaranteed. Algorithm 1 contains the pseudo-code for our approach, while Appendix D details derivations and further properties of the linear system.

## 5. Theoretical Analysis

In this section, we provide a convergence analysis of our algorithm. We assume that the user and item indices, denoted by random variables $L$ and $I$, respectively, are independent and uniformly distributed as $L \sim \text{Uniform}([n])$ and $I \sim \text{Uniform}([m])$. Conditional on $(L, I)$, the observed response $X \sim \text{Bernoulli}(\sigma(\theta_L^d - \beta_I^d))$, as per the Rasch model, where $\phi^d$ are the true, data-generating, parameters. We further assume that the parameters are bounded as $|\theta_l| \leq 1$ for all $l$ and $|\beta_i| \leq 1$ for all $i$.

Up to this point, we have worked with the sample $\widetilde{Q}$ function. We now denote the sample objective by $\widetilde{Q}_N$ and derive the population $\widetilde{Q}$ function, which will be used for theoretical analysis:

$$\widetilde{Q}(\phi \mid \phi^{(t)}) = -\frac{1}{2} \mathbb{E}_{L,I,X} \Big[ (-1)^X \cdot (\theta_L - \beta_I) \\ + \kappa\left(\eta_{LI}^{(t)}\right) \cdot (\theta_L - \beta_I)^2 \Big] \\ - \frac{\nu}{2} \|\phi\|_2^2$$

Following Balakrishnan et al. (2017), we first establish four key properties of the population $\widetilde{Q}$ function that form the basis of our convergence analysis. Let $\phi^* = \arg\max_{\phi} \widetilde{Q}(\phi \mid \phi^*)$, define function $\tilde{q} := \widetilde{Q}(\phi \mid \phi^*)$, and let $\mathcal{B}_2(r; \phi^*)$ denote the Euclidean ball of radius $r > 0$ centered at $\phi^*$. The properties can then be stated as:

**Property 1.** *[Gradient smoothness] For an appropriately small parameter $\gamma_S \geq 0$, we have that*

$$\|\nabla \tilde{q}(\phi) - \nabla \widetilde{Q}(\phi \mid \phi)\|_2 \leq \gamma_S \|\phi - \phi^*\|_2$$

*for all $\phi \in \mathcal{B}_2(r; \phi^*)$.*

**Property 2.** *[First-order stability (FOS)] For an appropriately small parameter $\gamma_F \geq 0$, we have that*

$$\|\nabla \widetilde{Q}(M(\phi) \mid \phi^*) - \nabla \widetilde{Q}(M(\phi) \mid \phi)\|_2 \leq \gamma_F \|\phi - \phi^*\|_2$$

*for all $\phi \in \mathcal{B}_2(r; \phi^*)$.*

**Property 3.** *[$\mu$-smoothness] There is some $\mu > 0$ such that*

$$\tilde{q}(\phi_1) - \tilde{q}(\phi_2) - \langle \nabla \tilde{q}(\phi_2), \phi_1 - \phi_2 \rangle \geq -\frac{\mu}{2} \|\phi_1 - \phi_2\|_2^2$$

*for all pairs $\phi_1, \phi_2 \in \mathcal{B}_2(r; \phi^*)$.*

**Property 4.** *[$\lambda$-strong concavity] There is some $\lambda > 0$ such that*

$$\tilde{q}(\phi_1) - \tilde{q}(\phi_2) - \langle \nabla \tilde{q}(\phi_2), \phi_1 - \phi_2 \rangle \leq -\frac{\lambda}{2} \|\phi_1 - \phi_2\|_2^2$$

*for all pairs $\phi_1, \phi_2 \in \mathcal{B}_2(r; \phi^*)$.*

We show that the population $\widetilde{Q}$ function satisfies all the four properties. Then, following Theorem 4 of Balakrishnan et al. (2017), we show the following local linear convergence guarantee for the population EM operator underlying our algorithm.

**Theorem 5.1.** *For some radius $r > 0$, parameter $\nu \geq 0.5$, and pair $\left(\frac{2}{5(4\nu-1)}, \nu\right)$, the function $\widetilde{Q}(\cdot \mid \phi^*)$ is globally $\nu$-strongly concave, and satisfies the FOS condition with parameter $\frac{2}{5(4\nu-1)}$ on the ball $\mathcal{B}_2(r; \phi^*)$. Then the population EM operator $M$ is contractive over $\mathcal{B}_2(r; \phi^*)$ with contraction coefficient $\rho = \frac{2}{5\nu(4\nu-1)} \leq 0.8$, that is,*

$$\|M(\phi) - \phi^*\|_2 \leq \rho \|\phi - \phi^*\|_2 \quad \text{for all } \phi \in \mathcal{B}_2(r; \phi^*).$$

*Further, for any initial point $\phi^{(0)} \in \mathcal{B}_2(r; \phi^*)$, the population EM sequence $\{\phi^{(t)}\}_{t=0}^{\infty}$ exhibits linear convergence, that is, for all $t = 1, 2, 3, \ldots,$*

$$\|\phi^{(t)} - \phi^*\|_2 \leq \rho^t \|\phi^{(0)} - \phi^*\|_2.$$

Likewise, following Theorem 1 of Balakrishnan et al. (2017), we obtain the following local linear convergence guarantee for a first-order variant of our algorithm.

**Theorem 5.2.** *For some radius $r > 0$, parameter $\nu > \frac{1}{5}$, and triplet $\left(\frac{1}{5}, \nu, \nu + \frac{1}{4}\right)$, the function $\tilde{q}(\phi) = \widetilde{Q}(\phi \mid \phi^*)$ is $\nu$-strongly concave, $\left(\nu + \frac{1}{4}\right)$-smooth, and satisfies the gradient smoothness condition with parameter $\frac{1}{5}$ on the ball $\mathcal{B}_2(r; \phi^*)$. Suppose the step-size is chosen as $\alpha = \frac{8}{8\nu+1}$. Then, for any initial point $\phi^{(0)} \in \mathcal{B}_2(r; \phi^*)$, the population gradient EM sequence $\{\phi^{(t)}\}_{t=0}^{\infty}$ exhibits linear convergence, that is, for all $t = 1, 2, 3, \ldots,$*

$$\|\phi^{(t)} - \phi^*\|_2 \leq \left(\frac{13}{40\nu + 5}\right)^t \|\phi^{(0)} - \phi^*\|_2.$$

Finally, Theorem 5 of Balakrishnan et al. (2017) extends the analysis to the finite-sample setting, giving the following guarantee.

**Theorem 5.3.** *Given the conditions of Theorem 5.1, for any initial point $\phi^{(0)} \in \mathcal{B}_2(r; \phi^*)$, if the sample size*

$$N \geq \frac{6K}{(1-\rho)r} \cdot \max\{n, m\} \cdot \ln\left(\frac{4\min\{n,m\}}{\delta}\right)$$

*for a sufficiently large constant $K > 0$, then the EM iterates $\{\phi^{(t)}\}_{t=0}^{\infty}$ satisfy the bound*

$$\|\phi^{(t)} - \phi^*\|_2 \leq \rho^t \|\phi^{(0)} - \phi^*\|_2$$
$$+ \frac{6K}{1-\rho} \cdot \frac{\max\{n, m\} \cdot \ln\left(\frac{4\min\{n,m\}}{\delta}\right)}{N}$$

*with probability at least $1 - \delta$.*

*Proof (Sketch).* Following Balakrishnan et al. (2017), we proceed by bounding the supremum of the $L_2$ norm of the deviation between the sample EM operator $M_N$ and the population EM operator $M$. We first reduce this operator-level bound to the maximum element-wise parameter differences,

$\max_l \left|\theta_l^{M_N} - \theta_l^{M}\right|$ and $\max_i \left|\beta_i^{M_N} - \beta_i^{M}\right|$. We then derive concentration bounds for the intermediate quantities appearing in the expressions for these differences, by applying Bernstein's inequality and Chernoff bounds along with appropriate union bounds. Substituting these concentration limits into the initial operator deviation bound and simplifying the expression gives the stated sample complexity guarantee with probability at least $1 - \delta$. $\square$

We refer the reader to Appendix E for proofs of the properties and theorems.

## 6. Experimental Analysis

In this section, we report experimental results on both synthetic and real-world datasets to evaluate the performance of our proposed PGEM algorithm against existing methods for estimating the Rasch model parameters.

### 6.1. Baselines

We compare our method to the following baselines: (1) JMLE, which performs joint maximum likelihood estimation of item and user parameters (Haberman, 2004), (2) MMLE, which uses a Bayesian approach with Gaussian priors on user parameters and performs marginal maximum likelihood estimation of item parameters (Bock & Aitkin, 1981), (3) CMLE, which estimates item parameters via conditional maximum likelihood estimation (Hambleton, 1991), and (4) Spectral method proposed by Nguyen & Zhang (2022) for estimation of item parameters.

As discussed earlier, methods (2) through (4) above only estimate the item parameters. The user parameters $\theta$ can be obtained via either maximum likelihood estimation (MLE) or maximum a posteriori (MAP) estimation. Since, the results using MLE for user parameters can be unstable in certain regimes, as discussed in Section 4.2, we use MAP estimation for user parameters in all our experiments for these baselines. This accounts to adding a regularization term to the likelihood objective for user parameters.

### 6.2. Datasets

**Synthetic data.** We generate synthetic data according to the Rasch model defined in Equation (2). The number of datapoints sampled depend on the density value $p$. The true item difficulties $\beta_j^*$ and user abilities $\theta_i^*$ are drawn independently from Gaussian distributions $\mathcal{N}(0, \sigma_\beta^2)$ and $\mathcal{N}(0, \sigma_\theta^2)$, respectively. Here, $m$ refers to number of items and $n$ refers to the number of users, $\theta^*$ and $\beta^*$ refer to true parameters for users and items whereas $\theta$ and $\beta$ refer to predicted user and item parameters, respectively. We report the normalized $l_2$ error. Experiments are carried over

20 random seeds, where each seed corresponds to a newly sampled set of $\theta^*$ and $\beta^*$. We consider the following two experimental settings:

1. **Varying number of users:** $m = 10, (\sigma_\beta, \sigma_\theta) = (1, 1), p = 0.05$, varying $n$ as $n \in \{50, 100, 500, 1000, 2500\}$.

2. **Varying sparsity level:** $m = 10$, $(\sigma_\beta, \sigma_\theta) = (1, 1), n = 100$, varying $p$ as $p \in \{0.01, 0.05, 0.1, 0.2, 0.5\}$.

**Real-world datasets.** We conduct experiments on multiple real-world datasets, ranging from small and medium-scale datasets such as UCI and ML-100K to large-scale datasets including BOOK-GENOME and ML-20M. Table 1 provides a summary of dataset sizes $(m, n)$ and density levels. We observe that in large-scale real-world datasets, the density is typically below 10%, and can be as low as $0.54\%$ in the case of ML-20M.

For preprocessing, we follow Nguyen & Zhang (2022) to convert ratings into binary labels. In the UCI Student dataset, grades labeled "Best" and "Vg" are mapped to 1, while all other grades are mapped to 0. In the 3-GRADES dataset, each grade is binarized independently by assigning labels above the median grade to 1 and the remaining labels to 0. For recommendation datasets such as MovieLens and BOOK-GENOME, ratings are binarized on a per-user basis: a rating is assigned label 1 if it exceeds the user's mean rating, and 0 otherwise. The ICAR-SAPA and MTurk-Lexical datasets were already preprocessed from source.

In the case of real-world datasets, one cannot directly observe the true values of these parameters, and has to rely on a separate test dataset for evaluation. Hence, we adopt the data-partitioning strategy proposed by Chen et al. (2019) which preserves identifiability of both student ability and item difficulty parameters.

**Data-partitioning strategy.** The strategy is described as follows: responses are split with an overall 80–20 train–test ratio. The procedure is implemented in two stages to satisfy standard IRT validity constraints. First, for each student, $80\%$ of their responses are assigned to the training set and the remaining $20\%$ to the test set, ensuring that every student appears at least once in training. Second, to correct for items that may be absent after student-wise stratification, any such item is reassigned by allocating $80\%$ of its responses to training and $20\%$ to testing. Although this correction may slightly reduce the effective test set size, it guarantees that every item appears in the training set. As a result, both ability parameters $\theta_j$ and difficulty parameters $\beta_i$ remain jointly estimable. This response-level stratified splitting strategy is therefore well suited for traditional IRT models

that estimate student and item parameters simultaneously, whereas student-level splits, such as those used in spectral methods like Nguyen & Zhang (2022), are compatible only with item-centric evaluation. For MMLE, we hold out 10% of the train split for validation using the same strategy.

*Table 1.* Dataset statistics. Here $n$ and $m$ denote the number of users and items, respectively. **Density (%)** is defined as the proportion of valid user–item responses relative to all possible $n \times m$ interactions.

| Dataset | Items ($m$) | Users ($n$) | Density (%) |
|---|---|---|---|
| ICAR-SAPA | 60 | 95236 | 20.99 |
| MTurk-Lexical | 4588 | 142 | 10.44 |
| ML-100K | 1682 | 943 | 6.30 |
| HETREC | 10109 | 2113 | 4.01 |
| ML-1M | 3706 | 6040 | 4.47 |
| ML-10M | 10677 | 69878 | 1.34 |
| BOOK-GENOME | 9374 | 350332 | 0.16 |
| ML-20M | 26744 | 138493 | 0.54 |
| UCI | 4 | 131 | 100.00 |
| 3 GRADES | 3 | 382 | 100.00 |

### 6.3. Evaluation Metrics

For synthetic data, since the true parameters are known, we evaluate estimation accuracy using the Root Mean Square Error (RMSE) between the estimated and true parameters for both item difficulties $\beta$ and user abilities $\theta$.

For real-world datasets, where the true parameters are not available, we assess model performance using the following metrics computed on a held-out test set. Let $\mathcal{D}_{\text{test}}$ denote the test dataset, consisting of tuples $(l, i)$ representing task-user interactions, and let $X_{li} \in \{0, 1\}$ denote the binary response of user $i$ to item $j$. Suppose that an algorithm learns parameters $\theta$ and $\beta$. The evaluation metrics measure the overall "agreement" of the predicted Rasch model probabilities $P_{li} = \sigma(\theta_l - \beta_i)$ with the observations $X_{li}$:

**Log-Likelihood.** The log-likelihood measures how well the model explains the observed responses under a Bernoulli assumption:

$$\mathcal{L} = \sum_{(l,i) \in \mathcal{D}_{\text{test}}} [X_{li} \log P_{li} + (1 - X_{li}) \log(1 - P_{li})]$$

This metric is directly aligned with maximum likelihood estimation and strongly penalizes overconfident incorrect predictions.

**Accuracy.** Accuracy evaluates the fraction of correctly classified responses in a 'stochastic sense' using a fixed decision threshold:

$$\text{Accuracy} = \frac{1}{|\mathcal{D}_{\text{test}}|} \sum_{(l,i) \in \mathcal{D}_{\text{test}}} \mathbb{I}\Big[(P_{li} > 0.5 \wedge X_{li} = 1)$$
$$\vee \ (P_{li} \leq 0.5 \wedge X_{li} = 0)\Big]$$

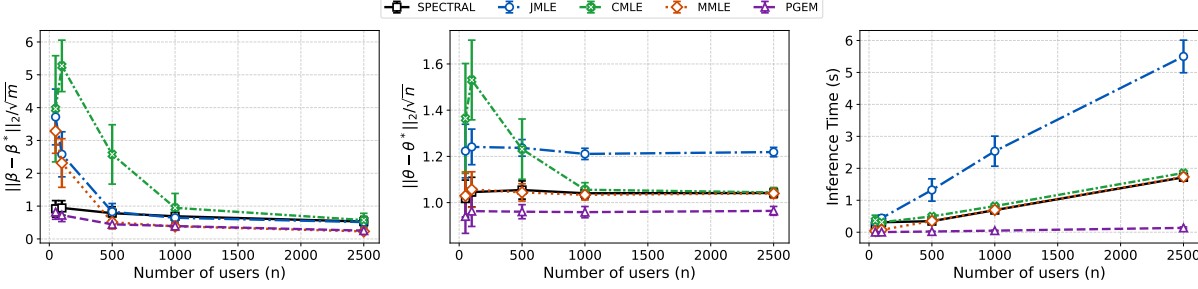

*(a)* Varying number of users ($m = 10, p = 0.05, n \in \{50, 100, 500, 1000, 2500\}, (\sigma_\beta, \sigma_\theta) = (1.0, 1.0)$.)

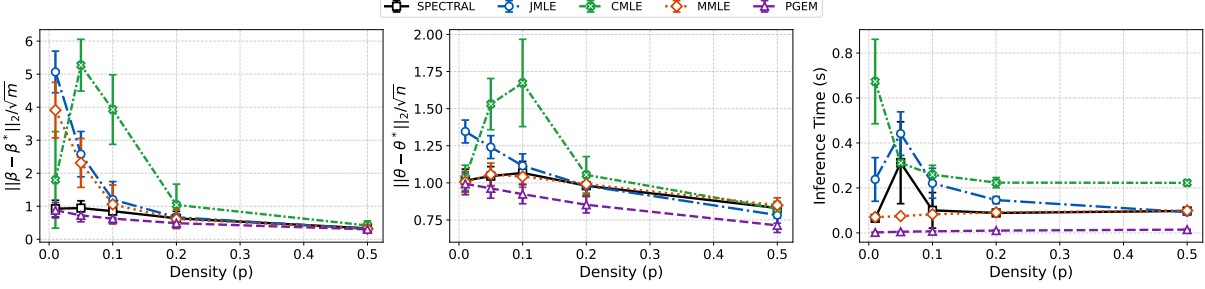

*(b)* Varying Sparsity: ($m = 10, p \in \{0.01, 0.05, 0.1, 0.2, 0.5\}, n = 100, (\sigma_\beta, \sigma_\theta) = (1.0, 1.0)$)).

*Figure 1.* Performance comparison of Rasch model estimation methods across synthetic experimental settings.

**Area Under the ROC Curve (AUC).** AUC measures the model's ability to rank positive responses higher than negative ones:

$$\text{AUC} = \frac{1}{|\mathcal{P}||\mathcal{N}|} \sum_{(l,i) \in \mathcal{P}} \sum_{(i',j') \in \mathcal{N}} \Big[ \mathbb{I}(P_{li} > P_{i'j'}) \\ + \tfrac{1}{2} \mathbb{I}(P_{li} = P_{i'j'}) \Big]$$

where $\mathcal{P}$ and $\mathcal{N}$ denote the sets of positive and negative responses, respectively. AUC is threshold-independent and robust to class imbalance, but it evaluates ranking quality only and does not assess probability calibration.

**Brier Score.** The Brier score evaluates the mean squared error between predicted probabilities and observed binary outcomes:

$$\text{Brier} = \frac{1}{|\mathcal{D}_{\text{test}}|} \sum_{(l,i) \in \mathcal{D}_{\text{test}}} (P_{li} - X_{li})^2$$

The Brier score is a strictly proper scoring rule and therefore incentivizes accurate predictions; a *smaller* value indicates better performance.

### 6.4. Implementation Details

We conduct our computations using Python 3.13 on a server with dual Intel Xeon Silver 4216 CPUs. For PGEM, we use

a regularization parameter of $\nu = 1.0$ in all experiments. We set the maximum number of EM iterations to $100$ and use a convergence tolerance of $1e - 5$ based on the relative change in log-likelihood. In the M-step, the resulting linear system is solved using the Gauss-Seidel method with a single iteration per EM step. The ablation study for these hyperparameters is detailed in Appendix G.

For JMLE, MMLE, and CMLE, we use the implementations in the popular `girth` package for IRT [2] with default settings. For the Spectral method, we use the code provided by Nguyen & Zhang (2022) and use the default options.

### 6.5. Results and Discussion

Experiments on synthetic datasets (Figure 1a, Figure 1b) indicate that spectral methods and CMLE exhibit degraded performance and increased runtime as density decreases. The results show that CMLE exhibits severe performance degradation in sparse regimes, consistent with prior findings. Joint estimation approaches such as JMLE also demonstrate inferior accuracy as sparsity increases. Methods primarily designed for estimating item parameters, such as the spectral approach of Nguyen & Zhang (2022), perform poorly in highly sparse settings. In contrast, PGEM maintains strong estimation accuracy for item parameters while also achiev-

---

[2] https://github.com/eribean/girth

*Table 2.* Metric-wise comparison across original datasets. Best values are highlighted per dataset. For log-likelihood (LOGLIK), area under the curve (AUC), and accuracy (ACC), higher values are better, whereas for time and Brier score (BRIER), lower values are better.

| Method | ICAR-SAPA | | | | | MTurk-Lexical | | | | | ML-100K | | | | |
|---|---|---|---|---|---|---|---|---|---|---|---|---|---|---|---|
| | Time | AUC | LogLik | Acc | Brier | Time | AUC | LogLik | Acc | Brier | Time | AUC | LogLik | Acc | Brier |
| JMLE | 210.79 | 0.785 | -0.598 | 0.719 | 0.190 | 33.42 | 0.80 | -0.328 | 0.889 | 0.087 | 32.472 | 0.726 | -0.655 | 0.671 | 0.219 |
| MMLE | 845.40 | 0.79 | -0.604 | 0.721 | 0.192 | 51.57 | 0.784 | -0.383 | 0.883 | 0.092 | 22.964 | 0.73 | -0.627 | 0.675 | 0.211 |
| CMLE | > 25k | | NA | | | 6953.60 | 0.5495 | -0.6981 | 0.8288 | 0.1391 | 4407.6 | 0.475 | -1.260 | 0.477 | 0.377 |
| SPECTRAL | 2128.68 | 0.804 | -0.538 | 0.724 | 0.181 | 3.49 | 0.811 | -0.295 | 0.885 | 0.087 | 2.862 | 0.728 | -0.607 | 0.670 | 0.210 |
| **PGEM (ours)** | 52.71 | 0.814 | -0.524 | 0.734 | 0.176 | 3.46 | 0.831 | -0.279 | 0.891 | 0.082 | 1.180 | 0.739 | -0.598 | 0.681 | 0.205 |

| Method | HETREC | | | | | BOOK-GENOME | | | | | ML-20M | | | | |
|---|---|---|---|---|---|---|---|---|---|---|---|---|---|---|---|
| | Time | AUC | LogLik | Acc | Brier | Time | AUC | LogLik | Acc | Brier | Time | AUC | LogLik | Acc | Brier |
| JMLE | 139.743 | 0.746 | -0.623 | 0.691 | 0.208 | 31566.2 | 0.657 | -0.838 | 0.615 | 0.263 | 26191.6 | 0.724 | -0.664 | 0.670 | 0.221 |
| MMLE | 122.848 | 0.754 | -0.597 | 0.697 | 0.200 | 1509.21 | 0.688 | -0.638 | 0.641 | 0.223 | 1522.63 | 0.746 | -0.594 | 0.688 | 0.203 |
| CMLE | > 15k | | NA | | | | | NA | | | | | NA | | |
| SPECTRAL | 111.96 | 0.752 | -0.588 | 0.695 | 0.201 | 10630 | 0.676 | -0.644 | 0.633 | 0.226 | 53174.9 | 0.745 | -0.592 | 0.687 | 0.203 |
| **PGEM (ours)** | 24.69 | 0.759 | -0.578 | 0.700 | 0.197 | 8318.86 | 0.698 | -0.634 | 0.647 | 0.221 | 4223.38 | 0.747 | -0.590 | 0.689 | 0.202 |

ing favorable computational efficiency, as evidenced by the inference time comparisons.

In the experiment where we vary the sparsity level while fixing the number of users to 100, PGEM consistently outperforms all competing methods in estimating both $\beta$ and $\theta$, with particularly pronounced improvements in highly sparse regimes. In contrast, the performance of Spectral degrades as sparsity increases. These results demonstrate the advantage of joint estimation with PGEM in sparse settings, which are characteristic of real-world datasets.

Table 2 shows the results for real-world datasets across different evaluation metrics. We can observe that PGEM consistently achieves strong predictive performance across various datasets, consistently outperforming baselines across all metrics. It also scales efficiently, remaining the fastest feasible method on very large datasets where other methods either fail or are prohibitively slow.

CMLE requires approximately 4,000 seconds to complete on ML-100K, whereas alternative algorithms terminate in under 50 seconds. Correspondingly, its predictive performance on ML-100K is lower, which is expected given the low density of the dataset. On HETREC, CMLE requires over 15,000 seconds and still fails to complete, while other methods finish in under 300 seconds. This is a known limitation of conditional likelihood approaches (Haberman, 2004). Consequently, we did not evaluate CMLE on larger datasets.

While MMLE appears faster on some datasets (BOOK-GENOME, ML-20M), note that the tables report per-prior runtime: since MMLE is known to be sensitive to prior misspecification, we need to tune the prior. For this, we follow the same approach as Nguyen & Zhang (2022), that is, we evaluate 15 priors and select the one that achieves the highest log-likelihood on a validation set formed by holding out 10% of the training data. The times reported for MMLE are averaged per prior, so the effective runtime scales as:

$$\text{Total Time} \approx \#\text{priors} \times \text{avg time per prior},$$

making it higher than our method in practice. Furthermore, the predictive performance of MMLE is inferior to PGEM across all metrics. The Spectral method is also computationally expensive on large datasets, and its predictive accuracy lags behind PGEM, particularly in sparse settings.

We also report experiments on filtered versions of the datasets, where users and items with very few observations are removed to obtain denser datasets, following the preprocessing protocol of Nguyen & Zhang (2022) for a more direct comparison. The results are given in Table 12 in Appendix H.3. PGEM continues to achieve the best performance in this setting, further demonstrating its effectiveness.

## 7. Conclusion

In this paper, we proposed an efficient joint learning algorithm based on Pólya-Gamma augmentation for making log-likelihood tractable. We utilized the Expectation-Maximization framework for estimating user and item pa-

rameters in the Rasch model. We provided a rigorous convergence analysis showing linear convergence to the maximum a posteriori (MAP) solution and demonstrated through extensive experiments on synthetic and real-world datasets that PGEM consistently outperforms existing methods in both accuracy and computational efficiency, particularly in sparse regimes where classical approaches such as JMLE, CMLE, MMLE, and spectral methods degrade or become impractical. These results highlight the importance of joint parameter estimation and the limitations of item-only or multi-stage estimation pipelines in modern large-scale IRT settings. Given the broad applicability of the Rasch model, we anticipate that the proposed approach will be valuable across a wide range of domains where reliable and scalable estimation of latent user and item parameters is required.

Future work includes extending the proposed framework to polytomous Rasch models as well as more expressive item response models, such as the three-parameter logistic (3PL) models. In addition, the Pólya–Gamma augmentation framework is applicable to a broader class of logistic latent variable models, and exploring its use in related estimation problems constitutes a promising direction for future research.

## Impact Statement

This work introduces a computationally efficient and theoretically grounded method for jointly estimating user abilities and item difficulties under the Rasch model. Item Response Theory is widely used in educational assessment, psychological testing, recommender systems, and crowdsourcing, where reliable parameter estimation is essential. By enabling stable joint estimation in sparse and large-scale settings, the proposed approach improves robustness and scalability compared to existing methods.

Potential positive impacts include more accurate assessment of learner abilities, improved calibration of test items, and better modeling of user–item interactions in data-sparse environments. At the same time, as with other IRT-based methods, application in high-stakes contexts such as education or evaluation requires careful consideration. Overreliance on automated estimates without appropriate validation or domain expertise may lead to unintended or unfair outcomes. These concerns arise from the broader use of latent variable models rather than from this method specifically.

This work advances the methodological foundations of joint learning in IRT, with potential benefits across multiple domains, while leaving ethical and policy considerations to context-specific deployment and governance.

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

# A. Related Work

We first review prior research on Item Response Theory (IRT), followed by an overview of the Rasch model and recent computational approaches for its estimation. Item Response Theory (IRT) is a fundamental framework in psychometrics with broad applications in educational assessment, psychology, and large-scale testing systems (Bock & Zimowski, 1997; Cai et al., 2016). IRT models are typically categorized into unidimensional and multidimensional formulations. Unidimensional IRT assumes that responses are governed by a single latent ability parameter $\theta$, whereas multidimensional IRT models capture multiple latent traits. Although multidimensional models are more expressive, they introduce substantial computational and statistical challenges; consequently, most practical and methodological work focuses on unidimensional IRT models.

Unidimensional IRT models are broadly categorized into dichotomous and polytomous models based on the response space. Dichotomous models handle binary responses (e.g., correct/incorrect), whereas polytomous models support multiple ordered or unordered response categories such as Likert-scale responses (Ostini & Nering, 2006; Nering & Ostini, 2010).

Dichotomous IRT models are commonly distinguished by the number of item parameters they employ (Thissen & Steinberg, 2009). Among dichotomous models, the Rasch or one-parameter logistic (1PL) model assumes a single item difficulty parameter with equal discrimination across items (Yang & Kao, 2014). The two-parameter logistic (2PL) model extends this formulation by incorporating item-specific discrimination parameters (Glas, 1999; Arifin & Yusoff, 2017; Jumini & Retnawati, 2022). More expressive variants include the three-parameter logistic (3PL) model, which accounts for guessing behavior (Lyu, 2023), and the four-parameter logistic (4PL) model, which additionally models upper-asymptote effects such as carelessness or slipping (Santoso et al., 2024).

In this work, we focus on the Rasch model (Rasch, 1980), a unidimensional dichotomous one-parameter logistic (1PL) model, along with its polytomous extensions (Andersen, 1995). Rasch models are among the most widely studied IRT formulations and have been applied across diverse domains (Bond & Fox, 2007).

Parameter estimation in Rasch models is commonly performed using maximum likelihood estimation (MLE) methods (Robitzsch, 2021; Linacre, 1999; 2004). Joint maximum likelihood estimation (JMLE) jointly estimates person and item parameters, but is known to exhibit consistency bias, particularly in small-sample settings. Marginal maximum likelihood estimation (MMLE) instead assumes a prior distribution over person abilities, typically Gaussian, and marginalizes them out when estimating item parameters. Although widely used in large-scale assessments (Zwinderman, 1991), MMLE can be sensitive to prior misspecification (Nguyen & Zhang, 2022). Conditional maximum likelihood estimation (CMLE) leverages sufficient statistics to estimate item parameters without explicitly modeling person abilities, but is known to perform poorly under sparse observations.

Recently, Nguyen & Zhang (2022) proposed a spectral approach for estimating item parameters of the Rasch model. The approach is comparatively faster and is robust to the issues that the prior maximum likelihood based techniques face. Building on this work, various studies have been developed applying this method. For example Nguyen & Zhang (2023) developed a private extension of the spectral algorithm which performs better in privacy preserving settings.

Despite the development of these methods, comparatively little emphasis has been placed on joint prediction of user and item parameters in the Rasch model. Classical approaches such as MMLE, CMLE, and spectral methods were originally designed to estimate only item parameters. Although user parameters can subsequently be inferred using likelihood-based techniques, both the evaluation protocols and the algorithmic formulations in these methods primarily focus on item parameter estimation. While JMLE jointly estimates user and item parameters, it is known to suffer from consistency issues and can be computationally expensive in large-scale settings. More recently, the spectral approach proposed by Nguyen & Zhang (2022) improves computational efficiency by reducing algorithmic complexity; however, the overall runtime remains considerable. Moreover, spectral approaches typically require high-density regimes, where most users provide responses for most items. Although such settings may arise in small-scale datasets, modern real-world datasets of practical significance are large-scale and highly sparse, with labeled user–item interactions often below 10%.

These limitations highlight the need for a computationally efficient joint learning framework that is robust in sparse regimes, achieves strong empirical performance in general, and avoids the fundamental issues associated with maximum likelihood–based techniques such as JMLE, MMLE, and CMLE.

## B. Pólya-Gamma Random Variables

Following Polson et al. (2013), if $\omega$ is a Pólya-Gamma distributed random variable with parameters $b > 0$ and $c \in \mathbb{R}$, denoted $\omega \sim \mathrm{PG}(b, c)$, then it is equal in distribution to an infinite weighted sum of gamma random variables:

$$\omega \overset{d}{=} \frac{1}{2\pi^2} \sum_{k=1}^{\infty} \frac{g_k}{\left(k - \frac{1}{2}\right)^2 + \frac{c^2}{4\pi^2}}, \quad g_k \sim \mathrm{Gamma}(b, 1).$$

Here, "$\overset{d}{=}$" denotes equality in distribution, and $g_k \sim \mathrm{Gamma}(b, 1)$ are independent Gamma random variables. Note that this is not the density function. Instead, equality in distribution means that the PG random variable on the left-hand side has the same cumulative distribution function as the random variable on the right-hand side.

(Polson et al., 2013) showed that all finite moments of $\omega$ admit closed form expressions; in particular,

$$\mathbb{E}[\omega] = \frac{b}{2c} \tanh\left(\frac{c}{2}\right). \tag{8}$$

They further proved the following two identities for $\omega \sim \mathrm{PG}(b, 0)$, which are central to Pólya-Gamma data augmentation for logistic models:

1. For any real $\psi$, one has

$$\frac{(e^\psi)^a}{(1 + e^\psi)^b} = 2^{-b} e^{\kappa\psi} \int_0^\infty e^{-\omega\psi^2/2} \, p(\omega \mid b, 0) \, d\omega, \tag{9}$$

   where $\kappa = a - \frac{b}{2}$, and $p(\omega \mid b, 0)$ is the density of a $\mathrm{PG}(b, 0)$ random variable.

2. The conditional density of $\omega$ given $\psi$ is again Pólya–Gamma with parameters $b$ and $\psi$, that is,

$$(\omega \mid \psi) \sim \mathrm{PG}(b, \psi).$$

## C. Pólya–Gamma ($\omega$) Augmentation and Log-Likelihood Derivation

Consider Equation (9). Let $\eta_{li} = \theta_l - \beta_i$. Then, for $a = 0$, $b = 1$, and $\psi = \eta_{li}$, we have $\kappa = -\frac{1}{2}$ and $p(\omega_{li}) = \mathrm{PG}(\omega_{li} \mid 1, 0)$, which allows us to write

$$\frac{1}{1 + e^{\eta_{li}}} = \frac{1}{2} e^{-\eta_{li}/2} \int_0^\infty \exp\left(-\frac{\omega_{li} \, \eta_{li}^2}{2}\right) p(\omega_{li}) \, d\omega_{li}.$$

Consequently, referring to Equation (3), we can write the likelihood contribution of the $(l, i)$-th response as

$$\begin{aligned}
p(X_{li} \mid \boldsymbol{\phi}) &= \frac{1}{1 + \exp\left\{(-1)^{X_{li}}(\theta_l - \beta_i)\right\}} \\
&= \frac{1}{2} \exp\left(-\frac{(-1)^{X_{li}} \eta_{li}}{2}\right) \int_0^\infty \exp\left(-\frac{\omega_{li}\eta_{li}^2 \cdot (-1)^{2X_{li}}}{2}\right) p(\omega_{li}) \, d\omega_{li} \\
&= \frac{1}{2} \int_0^\infty \exp\left(-\frac{(-1)^{X_{li}}\eta_{li}}{2} - \frac{\omega_{li}\eta_{li}^2}{2}\right) p(\omega_{li}) \, d\omega_{li},
\end{aligned}$$

which induces the augmented joint density

$$p(X_{li}, \omega_{li} \mid \boldsymbol{\phi}) \propto \exp\left(-\frac{(-1)^{X_{li}}\eta_{li}}{2} - \frac{\omega_{li}\eta_{li}^2}{2}\right) p(\omega_{li}).$$

Taking the logarithm and ignoring additive constants that do not depend on $\boldsymbol{\phi}$, the per-term log-likelihood simplifies to

$$\log p(X_{li}, \omega_{li} \mid \psi) \propto -\frac{(-1)^{X_{li}}}{2}\eta_{li} - \frac{\omega_{li}}{2}\eta_{li}^2 = -\frac{(-1)^{X_{li}}}{2}(\theta_l - \beta_i) - \frac{\omega_{li}}{2}(\theta_l - \beta_i)^2,$$

Summing over all observed entries $\mathcal{O}$ gives the *complete observed data* log-likelihood, given all latent variables $\omega_{li}$,

$$\ell(\boldsymbol{\phi}) = -\sum_{(l,i)\in\mathcal{O}} \left[\frac{(-1)^{X_{li}}}{2}(\theta_l - \beta_i) + \frac{1}{2}\omega_{li}(\theta_l - \beta_i)^2\right], \tag{10}$$

up to additive constants independent of $\boldsymbol{\phi}$.

# D. Expectation-Maximization

## D.1. E-Step

Given the current parameter estimate $\phi^{(t)}$, the E-step computes

$$Q\left(\phi \mid \phi^{(t)}\right) = \mathbb{E}_{\omega \mid X, \phi^{(t)}} \left[\ell(\phi)\right] = - \sum_{(l,i) \in \mathcal{O}} \mathbb{E}_{\omega_{li} \mid \phi^{(t)}} \left[ \frac{(-1)^{X_{li}}}{2} \left(\theta_l - \beta_i\right) + \frac{1}{2} \omega_{li} \left(\theta_l - \beta_i\right)^2 \right]. \tag{11}$$

The only term depending on $\omega_{li}$ is $\frac{1}{2}\omega_{li}\left(\theta_l - \beta_i\right)^2$. Pulling the other terms outside the expectation, we define

$$\kappa_{li}^{(t)} := \mathbb{E}\left[\omega_{li} \mid \phi^{(t)}\right].$$

Then,

$$Q\left(\phi \mid \phi^{(t)}\right) = - \sum_{(l,i) \in \mathcal{O}} \left[ \frac{(-1)^{X_{li}}}{2} \left(\theta_l - \beta_i\right) + \frac{1}{2} \kappa_{li}^{(t)} \left(\theta_l - \beta_i\right)^2 \right].$$

**Computing $\kappa_{li}^{(t)}$.** At iteration $t$, define

$$\eta_{li}^{(t)} = \theta_l^{(t)} - \beta_i^{(t)}.$$

Under the Pólya–Gamma augmentation, $\omega_{li} \mid \eta_{li}^{(t)} \sim \mathrm{PG}(1, \eta_{li}^{(t)})$. Its expectation (using Equation (8)) is

$$\kappa_{li}^{(t)} = \mathbb{E}\left[\omega_{li} \mid \phi^{(t)}\right] = \mathbb{E}\left[\omega_{li} \mid \eta_{li}^{(t)}\right] = \begin{cases} \frac{1}{2\,\eta_{li}^{(t)}} \tanh\left(\frac{\eta_{li}^{(t)}}{2}\right), & \eta_{li}^{(t)} \neq 0 \\ 1/4, & \eta_{li}^{(t)} = 0 \end{cases}$$

## D.2. M-Step

### D.2.1. UPDATING $\beta$

$$\widetilde{Q}(\phi \mid \phi^{(t)}) = -\frac{1}{|\mathcal{O}|} \sum_{(l,i) \in \mathcal{O}} \left[ \frac{(-1)^{X_{li}}}{2} \left(\theta_l - \beta_i\right) + \frac{1}{2} \kappa_{li}^{(t)} \left(\theta_l - \beta_i\right)^2 \right] - \frac{\nu}{2} \|\phi\|_2^2,$$

We first maximize $\widetilde{Q}$ with respect to each $\beta_i$. For a given item $i$, let $T_i = \{l \mid (l, i) \in \mathcal{O}\}$ denote the user index set. Then,

$$\frac{\partial \widetilde{Q}}{\partial \beta_i} = \frac{1}{|\mathcal{O}|} \sum_{l \in T_i} \left[ \frac{(-1)^{X_{li}}}{2} + \kappa_{li}^{(t)} \left(\theta_l - \beta_i\right) \right] - \nu \beta_i$$

For maximization, we require

$$\frac{\partial \widetilde{Q}}{\partial \beta_i} = 0,$$

i.e.,

$$\frac{1}{|\mathcal{O}|} \sum_{l \in T_i} \left[ \frac{(-1)^{X_{li}}}{2} + \kappa_{li}^{(t)} \left(\theta_l - \beta_i\right) \right] - \nu \beta_i = 0.$$

Rearrange to get the update equation:

$$\frac{1}{|\mathcal{O}|} \sum_{l \in T_i} \left( \kappa_{li}^{(t)} \theta_l + \frac{(-1)^{X_{li}}}{2} \right) - \beta_i \left( \nu + \frac{1}{|\mathcal{O}|} \sum_{l \in T_i} \kappa_{li}^{(t)} \right) = 0 \implies \beta_i = \frac{\frac{1}{|\mathcal{O}|} \sum_{l \in T_i} \left( \kappa_{li}^{(t)} \theta_l + \frac{(-1)^{X_{li}}}{2} \right)}{\nu + \frac{1}{|\mathcal{O}|} \sum_{l \in T_i} \kappa_{li}^{(t)}} \tag{12}$$

### D.2.2. UPDATING $\theta$

We now maximize $\widetilde{Q}$ with respect to each $\theta_l$. For a given user $l$, let $S_l = \{i \mid (l, i) \in \mathcal{O}\}$ denote the item index set. Then,

$$\frac{\partial \widetilde{Q}}{\partial \theta_l} = -\frac{1}{|\mathcal{O}|} \sum_{i \in S_l} \left[ \frac{(-1)^{X_{li}}}{2} + \kappa_{li}^{(t)} (\theta_l - \beta_i) \right] - \nu \theta_l$$

For maximization, we require $\frac{\partial \widetilde{Q}}{\partial \theta_l} = 0$, i.e.,

$$\frac{1}{|\mathcal{O}|} \sum_{i \in S_l} \left[ \frac{(-1)^{X_{li}}}{2} + \kappa_{li}^{(t)} (\theta_l - \beta_i) \right] + \nu \theta_l = 0.$$

Rearrange to get the update equation:

$$\frac{1}{|\mathcal{O}|} \sum_{i \in S_l} \left( \frac{(-1)^{X_{li}}}{2} - \kappa_{li}^{(t)} \beta_i \right) + \theta_l \left( \nu + \frac{1}{|\mathcal{O}|} \sum_{i \in S_l} \kappa_{li}^{(t)} \right) = 0 \implies \theta_l = \frac{\frac{1}{|\mathcal{O}|} \sum_{i \in S_l} \left( \kappa_{li}^{(t)} \beta_i - \frac{(-1)^{X_{li}}}{2} \right)}{\nu + \frac{1}{|\mathcal{O}|} \sum_{i \in S_l} \kappa_{li}^{(t)}} \tag{13}$$

### D.2.3. LINEAR SYSTEM

In this section, we let $c_{li} := \kappa_{li}^{(t)}$ for notational convenience and define

$$\phi = \begin{bmatrix} \boldsymbol{\theta} \\ \boldsymbol{\beta} \end{bmatrix} \in \mathbb{R}^{n+m}, \quad H = \begin{bmatrix} H_{\boldsymbol{\theta}} & W \\ V & H_{\boldsymbol{\beta}} \end{bmatrix} \in \mathbb{R}^{(n+m) \times (n+m)}, \quad \boldsymbol{r} = \begin{bmatrix} \boldsymbol{r}_\theta \\ \boldsymbol{r}_\beta \end{bmatrix} \in \mathbb{R}^{n+m},$$

where

$$H_\theta = \mathrm{diag} \left( \nu + \frac{1}{|\mathcal{O}|} \sum_{i \in S_1} c_{1i}, \nu + \frac{1}{|\mathcal{O}|} \sum_{i \in S_2} c_{2i}, \ldots, \nu + \frac{1}{|\mathcal{O}|} \sum_{i \in S_n} c_{ni} \right), \quad W_{li} = -\frac{1}{|\mathcal{O}|} c_{li} \, \mathbb{I}\{i \in S_l\},$$

$$H_\beta = \mathrm{diag} \left( \nu + \frac{1}{|\mathcal{O}|} \sum_{l \in T_1} c_{l1}, \nu + \frac{1}{|\mathcal{O}|} \sum_{l \in T_2} c_{l2}, \ldots, \nu + \frac{1}{|\mathcal{O}|} \sum_{l \in T_m} c_{lm} \right), \quad V_{il} = -\frac{1}{|\mathcal{O}|} c_{li} \, \mathbb{I}\{l \in T_i\},$$

$$(\boldsymbol{r}_\theta)_l = -\frac{1}{|\mathcal{O}|} \sum_{i \in S_l} \frac{(-1)^{X_{li}}}{2}, \quad \text{and} \quad (\boldsymbol{r}_\beta)_i = \frac{1}{|\mathcal{O}|} \sum_{l \in T_i} \frac{(-1)^{X_{li}}}{2},$$

$\mathbb{I}[\cdot]$ is the indicator function, and $H_{\boldsymbol{\theta}} \in \mathbb{R}^{n \times n}, W \in \mathbb{R}^{n \times m}, V \in \mathbb{R}^{m \times n}$, and $H_{\boldsymbol{\beta}} \in \mathbb{R}^{m \times m}$. Note that both $W_{li}$ and $V_{il}$ correspond to the observed response between user $l$ and item $i$. Therefore, $W_{li} = V_{il}$ for all $(l, i)$, and $V = W^\top$. As a result, $H$ can be written as

$$H = \begin{bmatrix} H_{\boldsymbol{\theta}} & W \\ W^\top & H_{\boldsymbol{\beta}} \end{bmatrix} \in \mathbb{R}^{(n+m) \times (n+m)}$$

The solution to $\nabla \widetilde{Q} = 0$ can then be written as the solution to the linear system $H\phi = \boldsymbol{r}$. Now we will check the consistency of our linear system.

Consider any vector $\phi = \begin{pmatrix} \boldsymbol{\theta} \\ \boldsymbol{\beta} \end{pmatrix}, \phi \neq \mathbf{0}$. Then,

$$
\begin{aligned}
\boldsymbol{\phi}^\top H \boldsymbol{\phi} &= \begin{pmatrix} \boldsymbol{\theta}^\top & \boldsymbol{\beta}^\top \end{pmatrix} \begin{pmatrix} H_{\boldsymbol{\theta}} & W \\ W^\top & H_{\boldsymbol{\beta}} \end{pmatrix} \begin{pmatrix} \boldsymbol{\theta} \\ \boldsymbol{\beta} \end{pmatrix} \\
&= \boldsymbol{\theta}^\top H_{\boldsymbol{\theta}} \boldsymbol{\theta} + \boldsymbol{\theta}^\top W \boldsymbol{\beta} + \boldsymbol{\beta}^\top W^\top \boldsymbol{\theta} + \boldsymbol{\beta}^\top H_{\boldsymbol{\beta}} \boldsymbol{\beta} \\
&= \sum_l \Big( \nu + \frac{1}{|\mathcal{O}|} \sum_{i \in S_l} c_{li} \Big) \theta_l^2 - \frac{1}{|\mathcal{O}|} \sum_l \sum_{i \in S_l} c_{li} \beta_i \theta_l \, \mathbb{I}[i \in S_l] \\
&\quad - \frac{1}{|\mathcal{O}|} \sum_i \sum_{l \in T_i} c_{li} \theta_l \beta_i \, \mathbb{I}[l \in T_i] + \sum_i \Big( \nu + \frac{1}{|\mathcal{O}|} \sum_{l \in T_i} c_{li} \Big) \beta_i^2 \\
&= \frac{1}{|\mathcal{O}|} \left( \sum_l \sum_{i \in S_l} c_{li} \theta_l^2 - \sum_l \sum_{i \in S_l} c_{li} \beta_i \theta_l - \sum_i \sum_{l \in T_i} c_{li} \theta_l \beta_i + \sum_i \sum_{l \in T_i} c_{li} \beta_i^2 \right) + \nu \left( \sum_l \theta_l^2 + \sum_i \beta_i^2 \right) \\
&= \frac{1}{|\mathcal{O}|} \sum_{(l,i) \in \mathcal{O}} c_{li} \left( \theta_l^2 + \beta_i^2 - 2\theta_l \beta_i \right) + \nu \|\boldsymbol{\phi}\|_2^2 \\
&= \frac{1}{|\mathcal{O}|} \sum_{(l,i) \in \mathcal{O}} c_{li} \left( \theta_l - \beta_i \right)^2 + \nu \|\boldsymbol{\phi}\|_2^2,
\end{aligned}
$$

where the fourth equality follows since the inner sums are over elements of $S_l$ and $T_i$ itself. Since $c_{li} > 0$ and $\nu > 0$, $\boldsymbol{\phi}^\top H \boldsymbol{\phi} > 0$. Further, note that $H$ is a symmetric matrix. Therefore, $H$ is positive definite, and the linear system has a unique solution.

## E. Convergence Analysis of PGEM Algorithm

Let $N = |\mathcal{O}|$ be the number of observed samples/entries. Then, $\widetilde{Q}_N$ is the sample objective function defined as

$$
\widetilde{Q}_N(\boldsymbol{\phi} \mid \boldsymbol{\phi}^{(t)}) = -\frac{1}{|\mathcal{O}|} \sum_{(l,i) \in \mathcal{O}} \left[ \frac{(-1)^{X_{li}}}{2} (\theta_l - \beta_i) + \frac{1}{2} \kappa_{li}^{(t)} (\theta_l - \beta_i)^2 \right] - \frac{\nu}{2} \|\boldsymbol{\phi}\|_2^2 . \tag{14}
$$

Now, we assume that the user and item indices, denoted by random variables $L$ and $I$, respectively, are independent and uniformly distributed as $L \sim \mathrm{Uniform}([n])$ and $I \sim \mathrm{Uniform}([m])$. Conditional on $(L, I)$, the observed response $X \sim \mathrm{Bernoulli}(\sigma(\theta_L^d - \beta_I^d))$, as per the Rasch model, where $\boldsymbol{\phi}^d$ are the true, data-generating, parameters. We further assume that the parameters are bounded as $|\theta_l| \le 1$ for all $l$ and $|\beta_i| \le 1$ for all $i$.

The population $\widetilde{Q}$ function is defined as the limit of the sample $\widetilde{Q}_N$ function under an *infinite* sample size. Thus, it can be written as

$$
\widetilde{Q}(\boldsymbol{\phi} \mid \boldsymbol{\phi}^{(t)}) = -\frac{1}{2} \mathbb{E}_{L,I,X} \left[ (-1)^X \cdot (\theta_L - \beta_I) + \kappa \left( \eta_{LI}^{(t)} \right) \cdot (\theta_L - \beta_I)^2 \right] - \frac{\nu}{2} \|\boldsymbol{\phi}\|_2^2 ,
$$

and its gradient is given by

$$
\nabla \widetilde{Q}(\boldsymbol{\phi} \mid \boldsymbol{\phi}^{(t)}) = -\frac{1}{2} \mathbb{E}_{L,I,X} \left[ \left( (-1)^X + 2\kappa(\eta_{LI}^{(t)}) (\theta_L - \beta_I) \right) \begin{bmatrix} \boldsymbol{e_L} \\ -\boldsymbol{e_I} \end{bmatrix} \right] - \nu \boldsymbol{\phi}. \tag{15}
$$

Further, define the function $\tilde{q}$ by

$$
\tilde{q}(\boldsymbol{\phi}) := \widetilde{Q}(\boldsymbol{\phi} \mid \boldsymbol{\phi}^*)
$$

We will now show that the population objective $\widetilde{Q}$ satisfies Properties 1-4. Before presenting the properties, we first present a lemma which will help us in the subsequent proofs.

**Lemma 1.** *The function* $f(x) = \frac{1}{2x} \tanh\left(\frac{x}{2}\right)$, $f(0) = \frac{1}{4}$ *is Lipschitz continuous on* $\mathbb{R}$ *with Lipschitz constant* $\lambda_0 = \frac{1}{20}$, *that is, for all* $x, x^* \in \mathbb{R}$,

$$
|f(x) - f(x^*)| \le \frac{1}{20} |x - x^*|
$$

*Proof.* We prove that $\sup_{x \in \mathbb{R}} |f'(x)| \leq \lambda_0 = \frac{1}{20}$, which will show that $f$ is Lipschitz continuous with constant $\lambda_0$.

For $x \neq 0$, $f'$ is given by the expression

$$f'(x) = \frac{x \, \text{sech}^2\left(\frac{x}{2}\right) - 2\tanh\left(\frac{x}{2}\right)}{4x^2},$$

and $f'(0) = 0$ is defined by $\lim_{x \to 0} f'(x)$.

Observe that $f'(x) \to 0$ as $|x| \to \infty$. As a result, $\sup_{x \in \mathbb{R}} |f'(x)|$ is attained on a bounded interval. Using the fact that $|\tanh(u)| \leq 1$ and $\text{sech}^2(u) \leq 1$, we can verify that $|f'(x)| \leq 0.01$ for $|x| \geq 10$. It therefore suffices to bound $|f'(x)|$ on the interval $[-10, 10]$.

Numerical maximization of $|f'(x)|$ on this interval confirms that $|f'(x)| \leq 0.043$ for all $x \in [-10, 10]$. Therefore, $|f'(x)| \leq 0.043 < \frac{1}{20}$ everywhere, which establishes the claimed Lipschitz constant. $\qquad\square$

We are now ready to prove the properties for the population objective.

**Property 1.** *[Gradient smoothness] For an appropriately small parameter $\gamma_S \geq 0$, we have that*

$$\|\nabla \tilde{q}(\phi) - \nabla \widetilde{Q}(\phi \mid \phi)\|_2 \leq \gamma_S \|\phi - \phi^*\|_2$$

*for all $\phi \in \mathcal{B}_2(r; \phi^*)$.*

*Proof.* From Equation (15), we get that

$$\nabla \widetilde{Q}(\phi \mid \phi^*) - \nabla \widetilde{Q}(\phi \mid \phi) = -\mathbb{E}_{L,I}\left[(\kappa(\eta_{LI}^*) - \kappa(\eta_{LI}))(\theta_L - \beta_I)\begin{bmatrix} e_L \\ -e_I \end{bmatrix}\right]$$

Taking the Euclidean norm of this, we get

$$\begin{aligned}
\|\nabla \widetilde{Q}(\phi \mid \phi^*) - \nabla \widetilde{Q}(\phi \mid \phi)\|_2 &= \left\| \mathbb{E}_{L,I}\left[(\kappa(\eta_{LI}^*) - \kappa(\eta_{LI}))(\theta_L - \beta_I)\begin{bmatrix} e_L \\ -e_I \end{bmatrix}\right] \right\|_2 \\
&\leq \mathbb{E}_{L,I}\left\| (\kappa(\eta_{LI}^*) - \kappa(\eta_{LI}))(\theta_L - \beta_I)\begin{bmatrix} e_L \\ -e_I \end{bmatrix} \right\|_2 \\
&= \mathbb{E}_{L,I}\left[|\kappa(\eta_{LI}^*) - \kappa(\eta_{LI})| \cdot |\theta_L - \beta_I| \cdot \left\| \begin{bmatrix} e_L \\ -e_I \end{bmatrix} \right\|_2\right]
\end{aligned} \tag{16}$$

Observe that the Euclidean norm of the random vector $\begin{bmatrix} e_L \\ -e_I \end{bmatrix}$ is deterministic and equals $\sqrt{2}$, since the vector has exactly two nonzero entries, one +1, one -1, with all others 0. Further, since $\theta_l$ and $\beta_i$ are assumed bounded above by 1, we have $|\theta_L - \beta_I| \leq |\theta_L| + |\beta_I| \leq 2$. Finally, by Lemma 1, the function $\kappa(\cdot)$ is Lipschitz continuous with Lipschitz constant $\lambda_0 = \frac{1}{20}$, which allows us to write,

$$|\kappa(\eta_{LI}^*) - \kappa(\eta_{LI})| \leq \lambda_0 \cdot |\eta_{LI}^* - \eta_{LI}| \leq \lambda_0 \cdot |(\theta_L^* - \beta_I^*) - (\theta_L - \beta_I)| \leq \lambda_0 \cdot |\theta_L^* - \theta_L| + \lambda_0 \cdot |\beta_I^* - \beta_I|$$

Substituting these in Equation (16), we get

$$\begin{aligned}
\|\nabla \widetilde{Q}(\phi \mid \phi^*) - \nabla \widetilde{Q}(\phi \mid \phi)\|_2 &\leq \mathbb{E}_{L,I}\left[|\kappa(\eta_{LI}^*) - \kappa(\eta_{LI})| \cdot |\theta_L - \beta_I| \cdot \left\| \begin{bmatrix} e_L \\ -e_I \end{bmatrix} \right\|_2\right] \\
&\leq \mathbb{E}_{L,I}\left[\lambda_0 \cdot (|\theta_L^* - \theta_L| + |\beta_I^* - \beta_I|) \cdot 2\sqrt{2}\right] \\
&\leq 2\sqrt{2}\,\lambda_0 \cdot (\mathbb{E}_L\,|\theta_L^* - \theta_L| + \mathbb{E}_I\,|\beta_I^* - \beta_I|)
\end{aligned} \tag{17}$$

Now, for any realization of $L$, $|\theta_L^* - \theta_L| \leq \|\theta^* - \theta\|_2$. Therefore, $\mathbb{E}_L\,|\theta_L^* - \theta_L| \leq \|\theta^* - \theta\|_2$, and similarly, $\mathbb{E}_I\,|\beta_I^* - \beta_I| \leq \|\beta^* - \beta\|_2$. This gives us

$$\|\nabla \widetilde{Q}(\phi \mid \phi^*) - \nabla \widetilde{Q}(\phi \mid \phi)\|_2 \leq 2\sqrt{2}\,\lambda_0 \cdot \sqrt{2}\|\phi^* - \phi\|_2 = \frac{1}{5}\|\phi^* - \phi\|_2$$

which completes the proof with $\gamma_{GS} = \frac{1}{5}$. $\qquad\square$

Before stating the next property, we define the operator $M$ as

$$M(\phi) := \arg\max_{\phi'} \widetilde{Q}(\phi' \mid \phi).$$

Using this notation allows us to write each iterate of the EM algorithm as $\phi^{(t+1)} = M(\phi^{(t)})$. Let $\theta_i^M$ and $\beta_i^M$ denote the elements of the vector $M(\phi)$.

**Property 2.** *[First-order stability (FOS)]  For an appropriately small parameter $\gamma_F \geq 0$, we have that*

$$\|\nabla \widetilde{Q}(M(\phi) \mid \phi^*) - \nabla \widetilde{Q}(M(\phi) \mid \phi)\|_2 \leq \gamma_F \|\phi - \phi^*\|_2$$

*for all $\phi \in \mathcal{B}_2(r; \phi^*)$.*

*Proof.*  From Equation (16), we get that

$$\|\nabla\widetilde{Q}(M(\phi) \mid \phi^*) - \nabla\widetilde{Q}\left(M(\phi) \mid \phi\right)\|_2 \leq \mathbb{E}_{L,I}\left[|\kappa(\eta_{LI}^*) - \kappa(\eta_{LI})| \cdot |\theta_L^M - \beta_I^M| \cdot \left\|\begin{bmatrix} e_L \\ -e_I \end{bmatrix}\right\|_2\right]. \qquad (18)$$

As shown in the previous proof, the Euclidean norm of the random vector $\begin{bmatrix} e_L \\ -e_I \end{bmatrix}$ equals $\sqrt{2}$, and by Lemma 1, we can write

$$|\kappa(\eta_{LI}^*) - \kappa(\eta_{LI})| \leq \lambda_0 \sqrt{2} \|\phi^* - \phi\|_2.$$

Substituting these in Equation (18), we get

$$\|\nabla\widetilde{Q}(M(\phi) \mid \phi^*) - \nabla\widetilde{Q}\left(M(\phi) \mid \phi\right)\|_2 \leq 2\lambda_0 \mathbb{E}_{L,I}\left|\theta_L^M - \beta_I^M\right| \cdot \|\phi^* - \phi\|_2 \qquad (19)$$

We now need to bound the term $\mathbb{E}_{L,I}\left|\theta_L^M - \beta_I^M\right|$. We first derive the expressions for elements of $M(\phi)$. Analogous to the derivation of the M-step updates of our algorithm in Appendix D.2, we can derive the expression for $M(\phi)$ updates for the population $\widetilde{Q}$ function as

$$\theta_l^M = \frac{\frac{1}{n}\mathbb{E}_I\left[\kappa_{lI} \cdot \beta_I^M\right] - \frac{1}{n}\mathbb{E}_X\left[\frac{(-1)^X}{2}\right]}{\nu + \frac{1}{n}\mathbb{E}_I\left[\kappa_{lI}\right]}, \quad \beta_i^M = \frac{\frac{1}{m}\mathbb{E}_L\left[\kappa_{Li} \cdot \theta_L^M\right] + \frac{1}{m}\mathbb{E}_X\left[\frac{(-1)^X}{2}\right]}{\nu + \frac{1}{m}\mathbb{E}_L\left[\kappa_{Li}\right]}. \qquad (20)$$

We now bound $|\theta_L^M|$ and $|\beta_I^M|$. Letting $B = \max_i |\beta_i^M|$ and $T = \max_l |\theta_l^M|$,

$$\left|\theta_l^M\right| \leq \frac{\frac{1}{n}\left|\mathbb{E}_I\left[\kappa_{lI} \cdot \beta_I^M\right]\right| + \frac{1}{n}\left|\frac{1}{2}\mathbb{E}_X\left[(-1)^X\right]\right|}{\nu + \frac{1}{n}\mathbb{E}_I\left[\kappa_{lI}\right]} \leq \frac{\mathbb{E}_I\left[|\kappa_{lI}| \cdot |\beta_I^M|\right] + \frac{1}{2}}{\nu} \qquad (21)$$

$$\leq \frac{\frac{1}{4}\left|\max_i \beta_i^M\right| + \frac{1}{2}}{\nu} \leq \frac{\frac{B}{4} + \frac{1}{2}}{\nu} = \frac{2 + B}{4\nu},$$

where the second inequality follows from the fact that $\left|\mathbb{E}_X\left[(-1)^X\right]\right| \leq \mathbb{E}_X\left|(-1)^X\right| \leq 1$, and the third inequality follows since $\kappa(\cdot) \leq \frac{1}{4}$. Since this inequality is satisfied for all $l$, we can say $T \leq \frac{2+B}{4\nu}$. Similarly, we can show that

$$\left|\beta_i^M\right| \leq \frac{2 + T}{4\nu},$$

for all $i$, which gives $B \leq \frac{2+T}{4\nu}$. Now, substituting this upper bound for $B$ in Equation (21), we get

$$T \leq \frac{2 + \frac{2+T}{4\nu}}{4\nu} \implies T \leq \frac{2(4\nu + 1)}{16\nu^2 - 1} = \frac{2}{4\nu - 1},$$

and similarly, $B \leq \frac{2}{4\nu-1}$. These expressions are valid for $\nu > \frac{1}{4}$.

Finally, with these bounds, Equation (19) can be written as:

$$
\begin{aligned}
\|\nabla \widetilde{Q}(M(\phi) \mid \phi^*) - \nabla \widetilde{Q}(M(\phi) \mid \phi)\|_2 &\leq 2\lambda_0 \mathbb{E}_{L,I} \left| \theta_L^M - \beta_I^M \right| \cdot \|\phi^* - \phi\|_2 \\
&\leq 2\lambda_0 \left( \left| \max_l \theta_l^M \right| + \left| \max_i \beta_i^M \right| \right) \cdot \|\phi^* - \phi\|_2 \\
&\leq 2\lambda_0 \left( T + B \right) \cdot \|\phi^* - \phi\|_2 \\
&\leq 2\lambda_0 \cdot \frac{4}{4\nu - 1} \cdot \|\phi^* - \phi\|_2 = \frac{2}{5\left(4\nu - 1\right)} \|\phi^* - \phi\|_2,
\end{aligned}
$$

which completes the proof with $\gamma_F = \frac{2}{5(4\nu - 1)}$. $\qquad\square$

**Property 3.** *[$\mu$-smoothness] There is some $\mu > 0$ such that*

$$
\tilde{q}(\phi_1) - \tilde{q}(\phi_2) - \langle \nabla \tilde{q}(\phi_2), \phi_1 - \phi_2 \rangle \geq -\frac{\mu}{2} \|\phi_1 - \phi_2\|_2^2
$$

*for all pairs $\phi_1, \phi_2 \in \mathcal{B}_2(r; \phi^*)$.*

*Proof.* In this proof and the next, we will denote elements of $\phi_1$ by $(\theta_{L,1}, \beta_{I,1})$, and elements of $\phi_2$ by $(\theta_{L,2}, \beta_{I,2})$. For notational convenience, we drop the indexing for $\eta_{LI}$, and define $\eta_1 := \theta_{L,1} - \beta_{I,1}$, corresponding to the vector $\phi_1$; $\eta_2 := \theta_{L,2} - \beta_{I,2}$, corresponding to the vector $\phi_2$; and $\eta^* := \theta_L^* - \beta_I^*$, corresponding to the vector $\phi^*$. With this notation, the $\tilde{q}$ function can be written as

$$
\tilde{q}(\phi) = \widetilde{Q}(\phi \mid \phi^*) = -\frac{1}{2} \mathbb{E}_{L,I,X} \left[ (-1)^X \cdot \eta + \kappa\left(\eta^*\right) \cdot \eta^2 \right] - \frac{\nu}{2} \|\phi\|_2^2
$$

We start by computing the LHS term as

$$
\begin{aligned}
\text{LHS} &= -\frac{1}{2} \mathbb{E}_{L,I,X} \left[ (-1)^X (\eta_1 - \eta_2) + \kappa\left(\eta^*\right)\left(\eta_1^2 - \eta_2^2\right) \right] - \frac{\nu}{2} \left( \|\phi_1\|_2^2 - \|\phi_2\|_2^2 \right) \\
&\quad + \frac{1}{2} (\phi_1 - \phi_2)^\top \mathbb{E}_{L,I,X} \left[ \left((-1)^X + 2\eta_2 \cdot \kappa\left(\eta^*\right)\right) \begin{pmatrix} e_L \\ -e_I \end{pmatrix} \right] + \nu (\phi_1 - \phi_2)^\top \phi_2 \qquad (22) \\
&= A_1 + A_2 + A_3 + A_4 \text{ (say)}
\end{aligned}
$$

We will first simplify the terms:

$$
\begin{aligned}
A_3 &= \frac{1}{2} \mathbb{E}_{L,I,X} \left[ \left((-1)^X + 2\kappa\left(\eta^*\right) \cdot \eta_2\right) \cdot (\phi_1 - \phi_2)^\top \begin{pmatrix} e_L \\ -e_I \end{pmatrix} \right] \\
&= \frac{1}{2} \mathbb{E}_{L,I,X} \left[ \left((-1)^X + 2\kappa\left(\eta^*\right) \cdot \eta_2\right) \cdot \left((\theta_{L,1} - \theta_{L,2}) - (\beta_{I,1} - \beta_{I,2})\right) \right] \\
&= \frac{1}{2} \mathbb{E}_{L,I,X} \left[ \left((-1)^X + 2\kappa\left(\eta^*\right) \cdot \eta_2\right) \cdot (\eta_1 - \eta_2) \right].
\end{aligned}
$$

Now,

$$
A_1 + A_3 = \frac{1}{2} \mathbb{E}_{L,I,X} \left[ \kappa\left(\eta^*\right) \cdot \left(2\eta_2(\eta_1 - \eta_2) - \eta_1^2 + \eta_2^2\right) \right] = -\frac{1}{2} \mathbb{E} \left[ \kappa\left(\eta^*\right) \cdot (\eta_1 - \eta_2)^2 \right],
$$

and,

$$
\begin{aligned}
A_2 + A_4 &= \frac{\nu}{2} \left( 2\phi_1^\top \phi_2 - 2\phi_2^\top \phi_2 - \phi_1^\top \phi_1 + \phi_2^\top \phi_2 \right) \\
&= -\frac{\nu}{2} (\phi_1 - \phi_2)^\top (\phi_1 - \phi_2) = -\frac{\nu}{2} \|\phi_1 - \phi_2\|_2^2 \qquad (23)
\end{aligned}
$$

We know that $0 \leq \kappa(\cdot) \leq \frac{1}{4}$, which gives us

$$
-\frac{1}{8} \mathbb{E}_{L,I} (\eta_1 - \eta_2)^2 \leq A_1 + A_3 \leq 0 \qquad (24)
$$

Evaluating the expectation,

$$
\begin{aligned}
\mathbb{E}_{L,I} (\eta_1 - \eta_2)^2 &= \mathbb{E}_{L,I} \left( (\theta_{L,1} - \theta_{L,2}) - (\beta_{I,1} - \beta_{I,2}) \right)^2 \\
&= \mathbb{E}_L (\theta_{L,1} - \theta_{L,2})^2 + \mathbb{E}_I (\beta_{I,1} - \beta_{I,2})^2 + 2\mathbb{E}_L (\theta_{L,1} - \theta_{L,2}) \cdot \mathbb{E}_I (\beta_{I,2} - \beta_{I,1}).
\end{aligned}
\tag{25}
$$

Note that $\mathbb{E}_L (\theta_{L,1} - \theta_{L,2})^2 = \frac{1}{n} \sum_{l=1}^{n} (\theta_{l,1} - \theta_{l,2})^2 \leq \|\boldsymbol{\theta}_2 - \boldsymbol{\theta}_1\|_2^2$. Similarly, $\mathbb{E}_I (\beta_{I,1} - \beta_{I,2})^2 \leq \|\boldsymbol{\beta}_2 - \boldsymbol{\beta}_1\|_2^2$.

Further, $\mathbb{E}_I (\beta_{I,2} - \beta_{I,1}) = \frac{1}{m} \sum_{i=1}^{m} (\beta_{i,2} - \beta_{i,1}) = 0$ because of our centering constraint $\sum_i \beta_i = 0$. Therefore, $\mathbb{E}_L (\theta_{L,1} - \theta_{L,2}) \cdot \mathbb{E}_I (\beta_{I,2} - \beta_{I,1}) = 0$ and Equation (25) simplifies to

$$
\mathbb{E}_{L,I} (\eta_1 - \eta_2)^2 = \mathbb{E}_L (\theta_{L,1} - \theta_{L,2})^2 + \mathbb{E}_I (\beta_{I,1} - \beta_{I,2})^2 \leq \|\boldsymbol{\theta}_2 - \boldsymbol{\theta}_1\|_2^2 + \|\boldsymbol{\beta}_2 - \boldsymbol{\beta}_1\|_2^2 = \|\boldsymbol{\phi}_2 - \boldsymbol{\phi}_1\|_2^2
$$

Substituting this in Equation (24), we get

$$
A_1 + A_3 \geq -\frac{1}{8} \|\boldsymbol{\phi}_1 - \boldsymbol{\phi}_2\|_2^2.
\tag{26}
$$

Substituting Equations (26) and (23) in Equation (22), we get

$$
\text{LHS} \geq -\frac{1}{8} \|\boldsymbol{\phi}_1 - \boldsymbol{\phi}_2\|_2^2 - \frac{\nu}{2} \|\boldsymbol{\phi}_1 - \boldsymbol{\phi}_2\|_2^2 = -\frac{(1 + 4\nu)}{8} \|\boldsymbol{\phi}_1 - \boldsymbol{\phi}_2\|_2^2,
$$

which completes the proof with $\mu = \frac{1+4\nu}{4}$. $\qquad\square$

**Property 4.** *[$\lambda$-strong concavity] There is some $\lambda > 0$ such that*

$$
\tilde{q}(\boldsymbol{\phi}_1) - \tilde{q}(\boldsymbol{\phi}_2) - \langle \nabla \tilde{q}(\boldsymbol{\phi}_2), \boldsymbol{\phi}_1 - \boldsymbol{\phi}_2 \rangle \leq -\frac{\lambda}{2} \|\boldsymbol{\phi}_1 - \boldsymbol{\phi}_2\|_2^2
$$

*for all pairs $\boldsymbol{\phi}_1, \boldsymbol{\phi}_2 \in \mathcal{B}_2(r; \boldsymbol{\phi}^*)$.*

*Proof.* Following the proof for Property 4, we know from Equation (26) that

$$
A_1 + A_3 \leq 0,
\tag{27}
$$

and

$$
A_2 + A_4 = -\frac{\nu}{2} \|\boldsymbol{\phi}_1 - \boldsymbol{\phi}_2\|_2^2.
\tag{28}
$$

This gives us the following result

$$
LHS \geq -\frac{\nu}{2} \|\boldsymbol{\phi}_2 - \boldsymbol{\phi}_1\|_2^2,
\tag{29}
$$

which completes the proof with $\lambda = \nu$. $\qquad\square$

Thus, we have shown that the population objective $\widetilde{Q}$ first-order stability (Property 2) with parameter $\gamma_F = \frac{2}{5(4\nu-1)}$ and satisfies $\lambda$-strong concavity (Property 4) with $\lambda = \nu$. When $\nu \geq 0.5$, $\gamma_F$ is well-defined and $\gamma_F < \lambda$. Now, we leverage the framework of Balakrishnan et al. (2017) to provide the following results.

First, following Theorem 4 in Balakrishnan et al. (2017), we can state Theorem 5.1.

**Theorem 5.1.** *For some radius $r > 0$, parameter $\nu \geq 0.5$, and pair $\left( \frac{2}{5(4\nu-1)}, \nu \right)$, the function $\widetilde{Q}(\cdot \mid \boldsymbol{\phi}^*)$ is globally $\nu$-strongly concave, and satisfies the FOS condition with parameter $\frac{2}{5(4\nu-1)}$ on the ball $\mathcal{B}_2(r; \boldsymbol{\phi}^*)$. Then the population EM operator $M$ is contractive over $\mathcal{B}_2(r; \boldsymbol{\phi}^*)$ with contraction coefficient $\rho = \frac{2}{5\nu(4\nu-1)} \leq 0.8$, that is,*

$$
\|M(\boldsymbol{\phi}) - \boldsymbol{\phi}^*\|_2 \leq \rho \|\boldsymbol{\phi} - \boldsymbol{\phi}^*\|_2 \quad \text{for all } \boldsymbol{\phi} \in \mathcal{B}_2(r; \boldsymbol{\phi}^*).
$$

*Further, for any initial point $\boldsymbol{\phi}^{(0)} \in \mathcal{B}_2(r; \boldsymbol{\phi}^*)$, the population EM sequence $\{\boldsymbol{\phi}^{(t)}\}_{t=0}^{\infty}$ exhibits linear convergence, that is, for all $t = 1, 2, 3, \ldots$,*

$$
\|\boldsymbol{\phi}^{(t)} - \boldsymbol{\phi}^*\|_2 \leq \rho^t \|\boldsymbol{\phi}^{(0)} - \boldsymbol{\phi}^*\|_2.
$$

Further, the function $\widetilde{Q}$ satisfies gradient smoothness (Property 1) with parameter $\gamma_S = \frac{1}{5}$ and $\mu$-smoothness (Property 3) with $\mu = \nu + \frac{1}{4}$. When $\nu > \frac{1}{5}$, these constants satisfy $0 \leq \gamma_S < \lambda < \mu$, which allows us to state Theorem 5.2 to establish linear convergence of a first-order variant of our EM algorithm, where the M-step is replaced by a single gradient ascent step on $\widetilde{Q}$. This result follows directly from Theorem 1 of Balakrishnan et al. (2017).

**Theorem 5.2.** *For some radius $r > 0$, parameter $\nu > \frac{1}{5}$, and triplet $\left(\frac{1}{5}, \nu, \nu + \frac{1}{4}\right)$, the function $\tilde{q}(\phi) = \widetilde{Q}(\phi \mid \phi^*)$ is $\nu$-strongly concave, $\left(\nu + \frac{1}{4}\right)$-smooth, and satisfies the gradient smoothness condition with parameter $\frac{1}{5}$ on the ball $\mathcal{B}_2(r; \phi^*)$. Suppose the step-size is chosen as $\alpha = \frac{8}{8\nu + 1}$. Then, for any initial point $\phi^{(0)} \in \mathcal{B}_2(r; \phi^*)$, the population gradient EM sequence $\{\phi^{(t)}\}_{t=0}^{\infty}$ exhibits linear convergence, that is, for all $t = 1, 2, 3, \ldots,$*

$$\|\phi^{(t)} - \phi^*\|_2 \leq \left(\frac{13}{40\nu + 5}\right)^t \|\phi^{(0)} - \phi^*\|_2.$$

Finally, following Theorem 5 of Balakrishnan et al. (2017), we can obtain the following guarantee for sample-based EM.

**Theorem 5.3.** *Given the conditions of Theorem 5.1, for any initial point $\phi^{(0)} \in \mathcal{B}_2(r; \phi^*)$, if the sample size*

$$N \geq \frac{6K}{(1 - \rho)r} \cdot \max\{n, m\} \cdot \ln\left(\frac{4\min\{n, m\}}{\delta}\right)$$

*for a sufficiently large constant $K > 0$, then the EM iterates $\{\phi^{(t)}\}_{t=0}^{\infty}$ satisfy the bound*

$$\|\phi^{(t)} - \phi^*\|_2 \leq \rho^t \|\phi^{(0)} - \phi^*\|_2 + \frac{6K}{1 - \rho} \cdot \frac{\max\{n, m\} \cdot \ln\left(\frac{4\min\{n, m\}}{\delta}\right)}{N}$$

*with probability at least $1 - \delta$.*

*Proof.* We have to show that

$$\sup_{\phi \in \mathcal{B}_2(r; \phi^*)} \|M_N(\phi) - M(\phi)\|_2 \leq \frac{6K}{N} \cdot \max\{n, m\} \cdot \ln\left(\frac{4\min\{n, m\}}{\delta}\right)$$

Note that we can write

$$\|M_N(\phi) - M(\phi)\|_2 \leq \sqrt{n + m} \cdot \max\left\{\max_l \left|\theta_l^{M_N} - \theta_l^M\right|, \max_i \left|\beta_i^{M_N} - \beta_i^M\right|\right\} \tag{30}$$

We will first bound the term $\max_l \left|\theta_l^{M_N} - \theta_l^M\right|$ from above. Then, from Equations (13), (8), and (20), we get that

$$\theta_l^{M_N} = \frac{\frac{1}{|\mathcal{O}|} \sum_{i \in S_l} \left(\kappa_{li} \beta_i^{M_N} - \frac{(-1)^{X_{li}}}{2}\right)}{\nu + \frac{1}{|\mathcal{O}|} \sum_{i \in S_l} \kappa_{li}}, \quad \beta_i^{M_N} = \frac{\frac{1}{|\mathcal{O}|} \sum_{l \in T_i} \left(\kappa_{li} \theta_l^{M_N} + \frac{(-1)^{X_{li}}}{2}\right)}{\nu + \frac{1}{|\mathcal{O}|} \sum_{l \in T_i} \kappa_{li}},$$

and,

$$\theta_l^M = \frac{\frac{1}{n} \mathbb{E}_I\left[\kappa_{lI} \cdot \beta_I^M\right] - \frac{1}{n} \mathbb{E}_X\left[\frac{(-1)^X}{2}\right]}{\nu + \frac{1}{n} \mathbb{E}_I\left[\kappa_{lI}\right]}, \quad \beta_i^M = \frac{\frac{1}{m} \mathbb{E}_L\left[\kappa_{Li} \cdot \theta_L^M\right] + \frac{1}{m} \mathbb{E}_X\left[\frac{(-1)^X}{2}\right]}{\nu + \frac{1}{m} \mathbb{E}_L\left[\kappa_{Li}\right]}.$$

For a given $l \in [n]$, say $\theta_l^{M_N} = \frac{C_N}{D_N}$ and $\theta_l^M = \frac{C}{D}$.

Then, $\theta_l^{M_N} - \theta_l^M = \frac{C_N}{D_N} - \frac{C}{D} = \frac{C_N D - C D_N}{D_N D} = \frac{C_N D - C D + C D - C D_N}{D_N D} = \frac{(C_N - C)D + (D - D_N)C}{D_N D}.$

Since $D \geq \nu$ and $D_N \geq \nu$, we can write

$$\left|\theta_l^{M_N} - \theta_l^M\right| \leq \frac{1}{\nu^2}\left(|C_N - C| \cdot |D| + |D_N - D| \cdot |C|\right).$$

Further, from Equation (21), we know that $|C| \leq \frac{2+B}{4}$, and $|D| \leq \nu + \mathbb{E}_I |\kappa_{lI}| \leq \nu + \frac{1}{4}$, which gives us

$$\left| \theta_l^{M_N} - \theta_l^M \right| \leq \frac{1}{\nu^2} \left( \left( \nu + \frac{1}{4} \right) \cdot |C_N - C| + \left( \frac{2+B}{4} \right) \cdot |D_N - D| \right). \tag{31}$$

We first evaluate the term $|D_N - D|$. Observe that $N = \sum_{i=1}^m \sum_{l=1}^n M_{li}$ and $\mathbb{E}_M[N] = nmp$ since each $M_{li} \sim$ Bernoulli$(p)$.

$$
\begin{aligned}
|D_N - D| &= \left| \frac{1}{N} \sum_{j \in S_l} \kappa_{lj} - \frac{1}{n} \mathbb{E}_I[\kappa_{lI}] \right| \\
&= \left| \frac{1}{N} \sum_{j=1}^m M_{lj}\kappa_{lj} - \frac{1}{nm} \sum_{j=1}^m \mathbb{E}_I[\kappa_{lI}] \frac{\mathbb{E}_{M,I}[M_{lI}]}{\mathbb{E}_{M,I}[M_{lI}]} \right| \\
&= \left| \frac{1}{N} \sum_{j=1}^m M_{lj}\kappa_{lj} - \frac{1}{nmp} \sum_{j=1}^m \mathbb{E}_{M,I}[\kappa_{lI}M_{lI}] \right| \\
&= \left| \sum_{j=1}^m \frac{M_{lj}\kappa_{lj}}{N} - \sum_{j=1}^m \frac{\mathbb{E}_{M,I}[\kappa_{lI}M_{lI}]}{\mathbb{E}_M[N]} \right| \\
&= \left| \sum_{j=1}^m \left( \frac{M_{lj}\kappa_{lj}}{N} - \frac{\mathbb{E}_{M,I}[\kappa_{lI}M_{lI}]}{\mathbb{E}_M[N]} \right) \right| \\
&= \left| \frac{\sum_{j=1}^m \left( M_{lj}\kappa_{lj}\mathbb{E}_M[N] - N \cdot \mathbb{E}_{M,I}[\kappa_{lI}M_{lI}] \right)}{N \cdot \mathbb{E}_M[N]} \right| \\
&\leq \frac{1}{N} \left| \sum_{j=1}^m \left( M_{lj}\kappa_{lj} - \mathbb{E}_{M,I}[\kappa_{lI}M_{lI}] \right) \right| + \frac{1}{4nN} |N - \mathbb{E}_M[N]|, 
\end{aligned}
\tag{32}
$$

where the final inequality follows since $\mathbb{E}_{M,I}[\kappa_{lI}M_{lI}] = \mathbb{E}_I[\kappa_{lI}\mathbb{E}_M[M_{lI}]] \leq \frac{p}{4}$. Define the random variables $Z_j = M_{lj}\kappa_{lj} - \mathbb{E}_{M,I}[\kappa_{lI}M_{lI}]$. Then, $Z_j$ are independent, $\mathbb{E}_{M,I}Z_j = 0$, with $|Z_j| \leq \frac{1}{2}$ and $\mathbb{E}_{M,I}Z_j^2 \leq \frac{1}{16}$. Using Bernstein's inequality then gives

$$\Pr\left( \left| \sum_{j=1}^m Z_j \right| \geq N\varepsilon_0 \right) \leq 2\exp\left( -\frac{N^2\varepsilon_0^2}{\frac{N\varepsilon_0}{3} + \frac{m}{8}} \right) \leq 2\exp\left( -\frac{N^2\varepsilon_0^2}{N\varepsilon_0 + m} \right) \leq \frac{\delta}{2}.$$

Note that we bound this probability by $\delta/2$ so that the union bound in Equation (30) would result in probability bounded by $\delta$. Now, observe that this term $|D_N - D|$ is defined for a fixed user $l$. Since we are interested in $\max_l \left| \theta_l^{M_N} - \theta_l^M \right|$, we must take a union bound over all $l \in [n]$. Letting $t = \ln\left( \frac{4n}{\delta} \right)$ for notational convenience, and writing the quadratic inequality in $\varepsilon_0$, we get that

$$\varepsilon_0 \geq \frac{t}{N} + \frac{\sqrt{mt}}{N}. \tag{33}$$

Similarly, we can use a Chernoff bound for the second term as

$$\Pr\left( |N - \mathbb{E}_M[N]| \geq \varepsilon_1 \mathbb{E}_M[N] \right) \leq 2\exp\left( -\frac{nmp\varepsilon_1^2}{3} \right) \leq \frac{\delta}{2},$$

which gives

$$\varepsilon_1 \geq \sqrt{\frac{3\ln(\frac{4}{\delta})}{nmp}} \implies \varepsilon_1 \geq \frac{1}{p}\sqrt{\frac{3t}{nm}}, \tag{34}$$

where we can choose the second value for $\varepsilon_1$ to simplify our calculations since $p \leq 1$ and $\ln(\frac{4}{\delta}) \leq t$. Substituting Equations (33) and (34) in Equation (32), we get that with probability at least $1 - \delta/2$,

$$|D_N - D| \leq \frac{t}{N} + \frac{\sqrt{mt}}{N} + \frac{1}{N}\sqrt{\frac{mt}{n}} = \varepsilon' \text{ (say)} \tag{35}$$

We now do a similar analysis for the term $|C_N - C|$.

$$|C_N - C| = \left| \frac{1}{N} \cdot \sum_{j \in S_l} \kappa_{lj}\beta_j^{M_N} - \frac{1}{n}\mathbb{E}_I[\kappa_{lI} \cdot \beta_I^M] \right| + \frac{1}{2}\left| \frac{1}{N} \cdot \sum_{j \in S_l}(-1)^{X_{lj}} - \frac{1}{n}\mathbb{E}_X\left[(-1)^X\right] \right| \tag{36}$$

$$= \left| \frac{1}{N} \cdot \sum_{j=1}^{m} M_{lj}\kappa_{lj}\beta_j^{M_N} - \frac{1}{nmp}\sum_{j=1}^{m}\mathbb{E}_{M,I}[\kappa_{lI}M_{lI}\beta_I^M] \right|$$

$$+ \frac{1}{2}\left| \frac{1}{N} \cdot \sum_{j=1}^{m}(-1)^{X_{lj}}M_{lj} - \frac{1}{nmp}\sum_{j=1}^{m}\mathbb{E}_{X,M,I}\left[(-1)^X M_{lI}\right] \right|$$

$$\leq \frac{1}{N}\left| \sum_{j=1}^{m}\left(M_{lj}\kappa_{lj}\beta_j^{M_N} - \mathbb{E}_{M,I}[\kappa_{lI}M_{lI}\beta_I^M]\right) \right| + \frac{B}{4nN}|N - E_M[N]|$$

$$+ \frac{1}{2N}\left| \sum_{j=1}^{m}\left((-1)^{X_{lj}}M_{lj} - \mathbb{E}_{X,M,I}\left[(-1)^X M_{lI}\right]\right) \right| + \frac{1}{2nN}|N - E_M[N]|$$

$$= A_1 + A_2 + A_3 + A_4 \text{ (say)},$$

where we have used the fact that $\mathbb{E}_{M,I}[\kappa_{lI}M_{lI}\beta_I^M] \leq \frac{Bp}{4}$ and $\mathbb{E}_{X,M,I}\left[(-1)^X M_{lI}\right] \leq 1 \cdot p = p$. Now, for the term $A_1$, define the random variables $Z_j = M_{lj}\kappa_{lj}\beta_j^{M_N} - \mathbb{E}_{M,I}[\kappa_{lI}M_{lI}\beta_I^M]$. Then, $Z_j$ are independent, $\mathbb{E}_{M,I}Z_j = 0$, with $|Z_j| \leq \frac{B}{2}$ and $\mathbb{E}_{M,I}Z_j^2 \leq \frac{B^2}{16}$. Using Bernstein's inequality then gives

$$\Pr\left( \left| \sum_{j=1}^{m}Z_j \right| \geq N\varepsilon_2 \right) \leq 2\exp\left( -\frac{N^2\varepsilon_2^2}{\frac{BN\varepsilon_2}{3} + \frac{B^2m}{8}} \right) \leq 2\exp\left( -\frac{N^2\varepsilon_2^2}{BN\varepsilon_2 + B^2m} \right) \leq \frac{\delta}{2},$$

which gives

$$\varepsilon_2 \geq B\left( \frac{t}{N} + \frac{\sqrt{mt}}{N} \right).$$

For the term $A_3$, define the random variables $Z_j = (-1)^{X_{lj}}M_{lj} - \mathbb{E}_{X,M,I}\left[(-1)^X M_{lI}\right]$. Then, $Z_j$ are independent, $\mathbb{E}_{X,M,I}Z_j = 0$, with $|Z_j| \leq 2$ and $\mathbb{E}_{X,M,I}Z_j^2 \leq 1$. Using Bernstein's inequality then gives

$$\Pr\left( \left| \sum_{j=1}^{m}Z_j \right| \geq 2N\varepsilon_3 \right) \leq 2\exp\left( -\frac{2N^2\varepsilon_3^2}{\frac{4N\varepsilon_3}{3} + m} \right) \leq 2\exp\left( -\frac{N^2\varepsilon_3^2}{N\varepsilon_3 + m} \right) \leq \frac{\delta}{2},$$

which gives

$$\varepsilon_3 \geq \frac{t}{N} + \frac{\sqrt{mt}}{N}.$$

Note that the Chernoff bound for terms $A_2$ and $A_4$ remains the same. Therefore, we get that with probability at least $1 - \delta/2$, Equation (36) can be written as

$$|C_N - C| \leq (B+1)\left( \frac{t}{N} + \frac{\sqrt{mt}}{N} + \frac{1}{N}\sqrt{\frac{mt}{n}} \right) = (B+1)\varepsilon'. \tag{37}$$

Substituting Equations (37) and (35) in Equation (31), we get that with probability at least $1 - \delta/2$,

$$\left|\theta_l^{M_N} - \theta_l^M\right| \leq \frac{1}{\nu^2} \left(\left(\nu + \frac{1}{4}\right) \cdot |C_N - C| + \left(\frac{2+B}{4}\right) \cdot |D_N - D|\right)$$
$$\frac{1}{\nu^2} \left(\left(\nu + \frac{1}{4}\right)(B+1)\varepsilon' + \left(\frac{2+B}{4}\right)\varepsilon'\right)$$

Therefore, we can say with probability at least $1 - \delta/2$ that

$$\max_l \left|\theta_l^{M_N} - \theta_l^M\right| \leq K'\varepsilon'$$

for some large enough constant $K'$. Likewise, we can show that for $\varepsilon'' = \frac{u}{N} + \frac{\sqrt{nu}}{N} + \frac{1}{N}\sqrt{\frac{nu}{m}}$, with $u = \ln\left(\frac{4m}{\delta}\right)$,

$$\max_i \left|\beta_i^{M_N} - \beta_i^M\right| \leq K''\varepsilon''$$

with probability at least $1 - \delta/2$, for some large enough constant $K''$. Substituting these relations in Equation (30), we get that with probability at least $1 - \delta$,

$$\|M_N(\phi) - M(\phi)\|_2 \leq \sqrt{n+m} \cdot \max\left\{K'\varepsilon', K''\varepsilon''\right\}$$
$$\leq K\sqrt{n+m} \cdot \frac{1}{N}\max\left\{t + \sqrt{mt} + \sqrt{\frac{mt}{n}}, u + \sqrt{nu} + \sqrt{\frac{nu}{m}}\right\}$$
$$\leq 3K\sqrt{n+m} \cdot \frac{1}{N}\max\left\{t\sqrt{m}, u\sqrt{n}\right\}$$
$$\leq \frac{6K}{N} \cdot \max\left\{\sqrt{n}, \sqrt{m}\right\} \cdot \max\left\{\sqrt{m}\ln\left(\frac{4n}{\delta}\right), \sqrt{n}\ln\left(\frac{4m}{\delta}\right)\right\}$$
$$\leq \frac{6K}{N} \cdot \max\left\{n, m\right\} \cdot \ln\left(\frac{4\min\{n,m\}}{\delta}\right),$$

where $K = \max\{K', K''\}$. To understand the final inequality, observe that the function $f(k) = \frac{\sqrt{k}}{\ln(4k/\delta)}$ is increasing for all $k \geq 2$ when $\delta < 1$, and for all $k \geq 1$ when $\delta \leq \frac{1}{2}$, which suffices for our setting. As a result, we get that when $n \geq m$,

$$\frac{\sqrt{n}}{\ln(4n/\delta)} \geq \frac{\sqrt{m}}{\ln(4m/\delta)} \implies \sqrt{n}\ln(4m/\delta) \geq \sqrt{m}\ln(4n/\delta),$$

and the argument for $m \geq n$ follows symmetrically. Finally, with probability at least $1 - \delta$, we can say that

$$\sup_{\phi \in \mathcal{B}_2(r;\phi^*)} \|M_N(\phi) - M(\phi)\|_2 \leq \frac{6K}{N} \cdot \max\left\{n, m\right\} \cdot \ln\left(\frac{4\min\{n,m\}}{\delta}\right)$$

When the right-hand side quantity is upper bounded by $(1-\rho)r$, we get

$$N \geq \frac{6K}{(1-\rho)r} \cdot \max\left\{n, m\right\} \cdot \ln\left(\frac{4\min\{n,m\}}{\delta}\right),$$

where $\rho = \frac{2}{5\nu(4\nu-1)}$, and the bound is defined for $\nu \geq 0.5$ (as required by the conditions of Theorem 5.1). $\square$

# F. Datasets References

*Table 3.* Dataset statistics for original datasets. $n$ and $m$ denote the number of users and items, respectively. Density (%) is the proportion of observed interactions relative to all $n \times m$ pairs.

| Dataset | Items ($m$) | Users ($n$) | Valid Entries | Density (%) | References |
|---|---|---|---|---|---|
| ICAR-SAPA | 60 | 95236 | 1199896 | 20.99 | (Condon & Revelle, 2016) |
| MTurk-Lexical | 4588 | 142 | 68004 | 10.44 | (Ratcliff & Hendrickson, 2021) |
| BOOK GENOME | 9374 | 350332 | 5152656 | 0.16 | (Kotkov et al., 2022) |
| HETREC | 10109 | 2113 | 855598 | 4.01 | (Cantador et al., 2011) |
| ML-100K | 1682 | 943 | 100000 | 6.30 | (Harper & Konstan, 2015) |
| ML-1M | 3706 | 6040 | 1000209 | 4.47 | (Harper & Konstan, 2015) |
| ML-10M | 10677 | 69878 | 10000054 | 1.34 | (Harper & Konstan, 2015) |
| ML-20M | 26744 | 138493 | 20000263 | 0.54 | (Harper & Konstan, 2015) |
| UCI | 4 | 131 | 524 | 100.00 | (Hussain et al., 2018) |
| 3 GRADES | 3 | 382 | 1146 | 100.00 | (Cortez & Silva, 2008) |

*Table 4.* Dataset statistics after filtering out users and items (NS) with very few interactions.

| Dataset | Items ($m$) | Users ($n$) | Valid Entries | Density (%) | References |
|---|---|---|---|---|---|
| BOOK GENOME (NS) | 9374 | 49784 | 3561231 | 0.76 | (Kotkov et al., 2022) |
| HETREC (NS) | 3261 | 2113 | 756594 | 10.98 | (Cantador et al., 2011) |
| ML-100K (NS) | 1152 | 943 | 97953 | 9.02 | (Harper & Konstan, 2015) |
| ML-1M (NS) | 2934 | 6040 | 993114 | 5.60 | (Harper & Konstan, 2015) |
| ML-10M (NS) | 8653 | 69878 | 9978175 | 1.65 | (Harper & Konstan, 2015) |
| ML-20M (NS) | 10524 | 96876 | 18697778 | 1.83 | (Harper & Konstan, 2015) |

## G. Ablations

In Section 6.4, we described the hyperparameter settings used in our algorithm. In this section, we present ablation studies examining the sensitivity of the model to these hyperparameters.

Table 5 reports the results of varying the regularization parameter $\nu$ on the ML-1M and ML-100K datasets. We observe that larger values of $\nu$ lead to faster convergence, while the predictive metrics remain largely stable.

*Table 5.* Ablation on regularization parameter ($\nu$)

| Dataset | $\nu$ | LogLik | Acc | AUC | Brier | Time |
|---|---|---|---|---|---|---|
| | 0.01 | -0.5802 | 0.6964 | 0.7594 | 0.1977 | 63.78 |
| | 0.05 | -0.5798 | 0.6965 | 0.7595 | 0.1977 | 63.97 |
| | 0.10 | -0.5796 | 0.6965 | 0.7595 | 0.1976 | 52.71 |
| ML-1M | 0.25 | -0.5792 | 0.6965 | 0.7596 | 0.1975 | 30.32 |
| | 0.50 | -0.5788 | 0.6965 | 0.7598 | 0.1975 | 22.02 |
| | 1.00 | **-0.5786** | **0.6966** | **0.7598** | **0.1974** | 18.79 |
| | 1.50 | **-0.5786** | **0.6966** | **0.7598** | **0.1974** | **17.49** |
| | 0.01 | -0.6084 | 0.6810 | 0.7362 | 0.2073 | 6.13 |
| | 0.05 | -0.6045 | 0.6809 | 0.7366 | 0.2069 | 4.38 |
| | 0.10 | -0.6028 | 0.6806 | 0.7370 | 0.2066 | 2.86 |
| ML-100K | 0.25 | -0.6006 | 0.6809 | 0.7378 | 0.2060 | 1.66 |
| | 0.50 | -0.5990 | **0.6811** | 0.7384 | 0.2056 | 1.24 |
| | 1.00 | -0.5978 | **0.6811** | **0.7388** | 0.2053 | 1.10 |
| | 1.50 | **-0.5976** | 0.6809 | **0.7388** | **0.2052** | **1.05** |

In Table 6, we show the ablation results for $GS_{\max\_iter}$ (maximum number of iterations of the Gauss-Seidel loop) on the HETREC, ML-100K, and ML-1M datasets. We can observe that metrics change marginally with more iterations, suggesting that a single iteration of the loop is sufficient to achieve strong performance, thereby motivating our choice of this parameter for the main experiments.s

*Table 6.* Ablation on Gauss-Seidel Max Iter parameter

| Dataset | $GS_{\max\_iter}$ | LogLik | Acc | AUC | Brier | Time |
|---|---|---|---|---|---|---|
| | 1 | -0.5782 | 0.6998 | 0.7587 | 0.1969 | **23.49** |
| | 10 | -0.5782 | 0.6998 | 0.7587 | 0.1969 | 28.92 |
| HETREC | 50 | -0.5782 | 0.6998 | 0.7587 | 0.1969 | 28.79 |
| | 100 | -0.5782 | 0.6998 | 0.7587 | 0.1969 | 28.71 |
| | 1000 | -0.5782 | 0.6998 | 0.7587 | 0.1969 | 28.91 |
| | 5000 | -0.5782 | 0.6998 | 0.7587 | 0.1969 | 28.76 |
| | 1 | -0.5978 | 0.6811 | 0.7388 | 0.2053 | **1.37** |
| | 10 | -0.5978 | 0.6811 | 0.7388 | 0.2053 | 2.22 |
| ML-100K | 50 | -0.5978 | 0.6811 | 0.7388 | 0.2053 | 2.20 |
| | 100 | -0.5978 | 0.6811 | 0.7388 | 0.2053 | 2.18 |
| | 1000 | -0.5978 | 0.6811 | 0.7388 | 0.2053 | 2.18 |
| | 5000 | -0.5978 | 0.6811 | 0.7388 | 0.2053 | 2.19 |
| | 1 | -0.5786 | 0.6966 | 0.7598 | 0.1974 | **17.88** |
| | 10 | -0.5786 | 0.6966 | 0.7598 | 0.1974 | 22.13 |
| ML-1M | 50 | -0.5786 | 0.6966 | 0.7598 | 0.1974 | 22.11 |
| | 100 | -0.5786 | 0.6966 | 0.7598 | 0.1974 | 22.11 |
| | 1000 | -0.5786 | 0.6966 | 0.7598 | 0.1974 | 22.16 |
| | 5000 | -0.5786 | 0.6966 | 0.7598 | 0.1974 | 22.02 |

Table 7 reports the number of iterations required for convergence across datasets when $\nu = 1.0$. These results indicate that setting the maximum number of EM iterations to 100 is a conservative choice.

*Table 7.* Iterations to convergence with $\nu = 1.0$

| Dataset | Number of Iterations |
|---|---|
| HETREC | 38 |
| BOOK GENOME | 52 |
| ML-100K | 17 |
| ML-1M | 29 |
| ML-10M | 44 |
| ML-20M | 51 |

*Table 8.* Ablation on $EM_{loglik\_tolerance}$ for ML-1M dataset

| $EM_{loglik\_tolerance}$ | loglik | acc | auc | brier | time |
|---|---|---|---|---|---|
| $1e-3$ | -0.5786 | 0.6966 | 0.7598 | 0.1974 | **11.22** |
| $1e-4$ | -0.5786 | 0.6966 | 0.7598 | 0.1974 | 15.02 |
| $1e-5$ | -0.5786 | 0.6966 | 0.7598 | 0.1974 | 19.00 |
| $1e-6$ | -0.5786 | 0.6966 | 0.7598 | 0.1974 | 22.67 |

*Table 9.* Ablation on $GS_{var\_tolerance}$ for ML-1M dataset

| $GS_{var\_tolerance}$ | loglik | acc | auc | brier | time |
|---|---|---|---|---|---|
| $1e-2$ | -0.5786 | 0.6966 | 0.7598 | 0.1974 | **20.32** |
| $1e-3$ | -0.5786 | 0.6966 | 0.7598 | 0.1974 | 22.50 |
| $1e-4$ | -0.5786 | 0.6966 | 0.7598 | 0.1974 | 25.37 |
| $1e-5$ | -0.5786 | 0.6966 | 0.7598 | 0.1974 | 29.12 |

For the convergence tolerances, Table 8 reports the ablation results for $EM_{loglik\_tolerance}$ on the ML-1M dataset, while Table 9 presents the corresponding results for $GS_{var\_tolerance}$. In both cases, metrics show negligible change across settings. We set $EM_{loglik\_tolerance} = 1e-5$ and $GS_{var\_tolerance} = 1e-5$ as a balanced choice for our experiments.

## H. Additional Experiments

### H.1. Comparison with IRT Baselines under Ranking Metrics

We also evaluate our method using ranking metrics which are commonly used in the recommendation systems literature, by adapting them to the IRT setting. For each user, items labeled as 1 in the test set are treated as relevant (ground truth), and all test items that are unseen during training are used for evaluation. We report results for $K \in \{1, 4, 8\}$, corresponding to strict, moderate, and less strict top-$K$ evaluation settings respectively. Table 10 shows the comparison of PGEM with our baselines under these metrics. We find that our method remains competitive under these metrics as well.

*Table 10.* Comparison with IRT baselines under ranking metrics with $K \in \{1, 4, 8\}$.

| Dataset | Metric | PGEM | SPECTRAL | MMLE | JMLE |
|---|---|---|---|---|---|
| ML-100K | Hit@K | **0.77**, **0.98**, 0.99 | 0.74, **0.98**, **1.00** | 0.73, **0.98**, 0.99 | 0.75, **0.98**, 0.99 |
| | Recall@K | **0.15**, **0.47**, **0.68** | **0.15**, 0.46, **0.68** | 0.14, **0.47**, **0.68** | **0.15**, **0.47**, **0.68** |
| | NDCG@K | **0.77**, **0.75**, **0.77** | 0.74, 0.73, 0.76 | 0.73, 0.74, **0.77** | 0.75, 0.74, **0.77** |
| HETREC | Hit@K | **0.85**, **0.99**, **1.00** | **0.85**, **0.99**, **1.00** | 0.83, **0.99**, **1.00** | 0.82, **0.99**, **1.00** |
| | Recall@K | **0.07**, **0.24**, **0.37** | **0.07**, **0.24**, **0.37** | **0.07**, **0.24**, **0.37** | **0.07**, **0.24**, **0.37** |
| | NDCG@K | **0.85**, **0.83**, **0.81** | **0.85**, 0.81, 0.80 | 0.83, 0.81, 0.80 | 0.82, 0.81, 0.80 |
| ML-1M | Hit@K | **0.80**, **0.98**, **1.00** | 0.79, **0.98**, **1.00** | **0.80**, **0.98**, **1.00** | **0.80**, **0.98**, **1.00** |
| | Recall@K | **0.13**, **0.39**, **0.60** | **0.13**, **0.39**, 0.59 | **0.13**, **0.39**, 0.59 | **0.13**, **0.39**, **0.60** |
| | NDCG@K | **0.80**, **0.77**, **0.78** | 0.79, 0.76, **0.78** | **0.80**, 0.76, **0.78** | **0.80**, **0.77**, **0.78** |
| ML-10M | Hit@K | **0.81**, **0.98**, **1.00** | 0.80, **0.98**, **1.00** | **0.81**, **0.98**, **1.00** | 0.80, **0.98**, **1.00** |
| | Recall@K | **0.15**, **0.46**, **0.67** | **0.15**, **0.46**, 0.66 | **0.15**, **0.46**, **0.67** | **0.15**, **0.46**, **0.67** |
| | NDCG@K | **0.81**, **0.78**, **0.80** | 0.80, **0.78**, **0.80** | **0.81**, **0.78**, **0.80** | 0.80, **0.78**, **0.80** |
| ML-20M | Hit@K | **0.80**, **0.98**, **1.00** | 0.79, **0.98**, **1.00** | **0.80**, **0.98**, **1.00** | 0.79, **0.98**, **1.00** |
| | Recall@K | **0.15**, **0.46**, **0.67** | **0.15**, **0.46**, **0.67** | **0.15**, **0.46**, **0.67** | **0.15**, **0.46**, **0.67** |
| | NDCG@K | **0.80**, **0.77**, **0.80** | 0.79, **0.77**, 0.79 | **0.80**, **0.77**, 0.79 | 0.79, **0.77**, 0.79 |
| BOOK GENOME | Hit@K | **0.83**, **0.99**, **1.00** | 0.82, **0.99**, **1.00** | 0.82, **0.99**, **1.00** | 0.82, **0.99**, **1.00** |
| | Recall@K | **0.60**, **0.88**, **0.95** | **0.60**, **0.88**, **0.95** | **0.60**, **0.88**, **0.95** | **0.60**, **0.88**, **0.95** |
| | NDCG@K | **0.83**, **0.88**, **0.90** | 0.82, **0.88**, **0.90** | 0.82, **0.88**, **0.90** | 0.82, 0.87, 0.89 |

## H.2. Experiments on other datasets

In addition to the results presented in Table 2 of Section 6, we report experimental results on additional datasets, namely ML-1M, ML-10M, UCI, and 3 GRADES below.

*Table 11.* Metric-wise comparison across original datasets. Best values per dataset and metric are highlighted in bold. For loglikelihood (LOGLIK), Area under the curve (AUC) and accuracy (ACC) higher is better, whereas for time and brier score (BRIER) lower is better.

| DATASET | METRIC | PGEM (OURS) | SPECTRAL | JMLE | MMLE | CMLE |
|---------|--------|-------------|----------|------|------|------|
| UCI | TIME | 0.011 | 0.122 | 0.074 | 0.161 | 0.186 |
| | AUC | 0.834 | 0.835 | 0.833 | 0.802 | 0.838 |
| | LOGLIK | -0.53 | -0.479 | -0.495 | -0.517 | -0.479 |
| | ACC | 0.809 | 0.809 | 0.779 | 0.802 | 0.802 |
| | BRIER | 0.179 | 0.153 | 0.159 | 0.166 | 0.152 |
| 3 GRADES | TIME | 0.03 | 0.356 | 0.186 | 0.394 | 0.409 |
| | AUC | 0.962 | 0.942 | 0.963 | 0.923 | 0.933 |
| | LOGLIK | -0.471 | -0.336 | -0.382 | -0.409 | -0.342 |
| | ACC | 0.914 | 0.914 | 0.914 | 0.89 | 0.914 |
| | BRIER | 0.143 | 0.096 | 0.107 | 0.087 | 0.118 |
| ML-1M | TIME | 19.141 | 36.219 | 93.795 | 54.252 | > 64 K |
| | AUC | 0.760 | 0.756 | 0.748 | 0.757 | |
| | LOGLIK | -0.579 | -0.583 | -0.615 | -0.585 | NA |
| | ACC | 0.697 | 0.694 | 0.687 | 0.695 | |
| | BRIER | 0.197 | 0.199 | 0.208 | 0.199 | |
| ML-10M | TIME | 758.987 | 2560.936 | 8568.026 | 319.840 | |
| | AUC | 0.755 | 0.752 | 0.733 | 0.754 | |
| | LOGLIK | -0.584 | -0.586 | -0.650 | -0.588 | NA |
| | ACC | 0.694 | 0.693 | 0.676 | 0.693 | |
| | BRIER | 0.199 | 0.201 | 0.217 | 0.200 | |

## H.3. Experiments on datasets by removing sparse labelings

We construct denser versions of the datasets by filtering out users and items with very few interactions following Nguyen & Zhang (2022).

A cutoff specifies the minimum number of observed ratings required for a user or item to be retained after preprocessing. Specifically, we discard items with fewer than a dataset-specific *rating cutoff*, and for certain datasets we additionally discard users with fewer than a dataset-specific *number of ratings*. After filtering, we binarize the ratings relative to each user's mean rating, as in the original experimental setup. We use the same response-level stratified 80-20 split strategy detailed in Section 6.

The cutoff values are the same as those used by Nguyen & Zhang (2022). For BOOK-GENOME and ML-20M, both item-level and user-level filtering are applied; for all other datasets, only item-level filtering is applied. For ML-100K, we use `MOVIES_RATINGS_CUTOFF_LOW = 10`, and for ML-1M, ML-10M, and HET-REC, we use `MOVIES_RATINGS_CUTOFF = 25`. For ML-20M, we additionally apply user-level filtering with `MOVIES_RATINGS_CUTOFF_ML_20M = 50` and `USERS_RATINGS_CUTOFF_ML_20M = 40`. For BOOK-GENOME, we set `BOOK_GENOME_RATINGS_CUTOFF = 50` and `USERS_RATINGS_CUTOFF_GENOME = 25`.

Table 12 presents the results of these experiments.

*Table 12.* Metric-wise comparison across datasets. We remove those movies and users which have less labelings i.e., neglect sparse labelings denoted by (NS). This experiment tests PGEM in a higher-density settings than the original datasets. Best values per dataset and metric are highlighted . The values for UCI and 3 Grades datasets are the same as that of Table 3 as there are fully dense datasets.

| DATASET | METRIC | PGEM | SPECTRAL | JMLE | MMLE | CMLE |
|---|---|---|---|---|---|---|
| ML-100K (NS) | TIME | 0.84 | 1.908 | 19.432 | 12.815 | 2436.405 |
| | AUC | 0.746 | 0.734 | 0.735 | 0.743 | 0.470 |
| | LOGLIK | -0.592 | -0.602 | -0.634 | -0.601 | -1.510 |
| | ACC | 0.688 | 0.676 | 0.678 | 0.685 | 0.474 |
| | BRIER | 0.203 | 0.207 | 0.213 | 0.205 | 0.403 |
| HETREC (NS) | TIME | 7.696 | 6.971 | 35.973 | 34.262 | NA |
| | AUC | 0.760 | 0.757 | 0.748 | 0.758 | |
| | LOGLIK | -0.574 | -0.579 | -0.608 | -0.576 | |
| | ACC | 0.704 | 0.701 | 0.694 | 0.703 | |
| | BRIER | 0.195 | 0.197 | 0.205 | 0.196 | |
| ML-1M (NS) | TIME | 16.117 | 18.239 | 73.678 | 32.943 | NA |
| | AUC | 0.761 | 0.758 | 0.749 | 0.759 | |
| | LOGLIK | -0.577 | -0.581 | -0.581 | -0.613 | |
| | ACC | 0.697 | 0.694 | 0.688 | 0.695 | |
| | BRIER | 0.197 | 0.198 | 0.198 | 0.207 | |
| ML-10M (NS) | TIME | 601.36 | 1157.41 | 40015.692 | 246.227 | NA |
| | AUC | 0.754 | 0.752 | 0.732 | 0.754 | |
| | LOGLIK | -0.584 | -0.587 | -0.650 | -0.587 | |
| | ACC | 0.694 | 0.691 | 0.676 | 0.693 | |
| | BRIER | 0.200 | 0.201 | 0.217 | 0.200 | |
| BOOK GENOME (NS) | TIME | 495.152 | 2041.85 | 32658.659 | 212.040 | NA |
| | AUC | 0.727 | 0.714 | 0.712 | 0.724 | |
| | LOGLIK | -0.603 | -0.612 | -0.637 | -0.609 | |
| | ACC | 0.671 | 0.662 | 0.662 | 0.669 | |
| | BRIER | 0.208 | 0.212 | 0.218 | 0.210 | |
| ML-20M (NS) | TIME | 1724.916 | 1855.227 | 3462.005 | 412.357 | NA |
| | AUC | 0.751 | 0.749 | 0.735 | 0.750 | |
| | LOGLIK | -0.586 | -0.588 | -0.629 | -0.589 | |
| | ACC | 0.692 | 0.690 | 0.679 | 0.691 | |
| | BRIER | 0.201 | 0.201 | 0.213 | 0.201 | |

