# OpenReview forum: "An Efficient Joint Learning Approach for Item Response Theory"
_ICML.cc/2026/Conference — ICML 2026 regular_

### Official Review · Reviewer_CwQL · 2026-03-12

**Soundness:** 3
**Presentation:** 3
**Significance:** 2
**Originality:** 2
**Overall Recommendation:** 4
**Confidence:** 3

**Summary:**

This paper studies the practical problem of fitting a Rasch (1PL) model when the response matrix is large and heavily missing. The key idea is to use Polya-Gamma augmentation to transform the logistic likelihood into a conditionally Gaussian form, which leads to an EM algorithm: the E-step is available in closed form, and the M-step reduces to solving a quadratic optimization problem. The authors also provide a convergence analysis for the population EM operator under standard local regularity conditions. Empirically, the paper compares the proposed method with standard Rasch estimators and a spectral baseline on both synthetic and several real-world datasets, with a particular focus on sparse, large-scale settings where computational efficiency is especially important.

**Compliance With Llm Reviewing Policy:**

Affirmed.

**Final Justification:**

Overall, I think this is a technically solid paper. My main concerns were about the convergence claim and some missing implementation details, and the rebuttal addressed these points clearly enough to increase my confidence in the paper. I have raised my score accordingly.

**Key Questions For Authors:**

1. To me the convergence result is stated for population EM under local conditions. Could the authors clarify this and make sure the claim is stated precisely?

2. The theoretical assumptions do not seem to fully match the experimental setup. Could the authors talk more about this gap?

3. Could the authors provide more implementation details, especially preprocessing and solver settings?

**Limitations:**

Yes. It would still be helpful to discuss fairness and privacy issues a bit more.

**Strengths And Weaknesses:**

**Strengths**

- The paper studies the practical problem of fitting the Rasch model in large and sparse settings.

- The proposed method is clean and easy to follow. In particular, using Polya-Gamma augmentation to derive an EM algorithm seems reasonable and computationally efficient.

- The experiments are quite comprehensive, covering both synthetic and real datasets, and the runtime results are useful.

**Weaknesses**

- My main concern is that the theory seems a bit overstated. The convergence result is for population EM under local conditions, so it should not be presented too strongly.

- Some parts of the experimental setup and implementation are still unclear.

---

> ### Author Rebuttal · Authors · 2026-03-31
>
> We thank the reviewer for the comments.
>
> > Could the authors provide more implementation details, especially preprocessing and solver settings?
>
> ### Preprocessing Details
>
> We follow Nguyen & Zhang (2023a). The binary labels for different datasets are assigned as follows:
>
> **Education datasets**
> - **UCI Student:** “Best”/“Vg” → 1; others → 0
> - **3 Grades:** for each grade, above median → 1; otherwise → 0
>
> **Book/Movie datasets**
> - Per user: rating above user mean → 1; otherwise → 0
>
> ---
>
> ### Solver Settings
>
> Our EM algorithm uses the Gauss–Seidel method to solve the M-step linear system.
>
> **Default settings (paper experiments):**
> - $EM_\text{max-iter}=100$
> - $\nu=0.25$
> - $GS_\text{max-iter}=1$
> - EM tolerance: $10^{-5}$
> - GS tolerance: $10^{-3}$
>
> > The theoretical assumptions do not seem to fully match the experimental setup. Could the authors talk more about this gap?
>
> We acknowledge the gap. Our convergence result adds an assumption on the regularization parameter $\nu\ge0.5$.
>
> While our experiments were originally conducted with $\nu=0.25$, we conducted an Ablation Study as detailed in the response to Reviewer EgYv titled "Ablation Studies for the PGEM Algorithm". The study shows that using $\nu\ge0.5$ slightly improves the results and leads to faster convergence.
>
> We re-ran the experiments setting $\nu=1.0$, keeping all other parameters unchanged. This makes the experimental setup consistent with the theoretical assumptions. The updated results with this setting are provided in response to Reviewer EgYv. We can observe slight improvements in metrics and significant improvement in inference time as it leads to faster convergence.
>
> We hope this resolves the gap between theoretical assumptions and experimental setup. We again thank the reviewer for the comment.
>
> > To me the convergence result is stated for population EM under local conditions. Could the authors clarify this and make sure the claim is stated precisely? My main concern is that the theory seems a bit overstated. The convergence result is for population EM under local conditions, so it should not be presented too strongly.
>
> The reviewer is correct that our convergence result is stated for population EM under local conditions. In the final version of the paper, we plan to restate lines 199-201 as:
>
> **Following Theorem 1 of Balakrishnan et al. (2014), we show the following local linear convergence guarantee for the population EM operator underlying our algorithm.**
>
> Based on your comment, we were able to generalize our results to a sample-level convergence result which holds under similar local conditions. We hope this will help clarify how the behavior can extend to the finite-sample setting as well. In the final version, we will clearly differentiate between the population and sample-level analyses.
>
> Following Theorem 3 of Balakrishnan et al. (2014), we can obtain the following guarantee for sample-based EM.
>
> ### Theorem (Sample Complexity)
> Given the conditions of Theorem 5.1 (in our paper), for any initial point $\boldsymbol{\phi}^{(0)} \in \mathcal{B}_2 (r; \boldsymbol{\phi}^*)$, if the sample size
>
> $$
> N \ge \frac{6K}{(1-\rho)r} \cdot \max\\{n,m\\} \cdot \ln\Big( \frac{4\min\\{n,m\\}}{\delta} \Big)
> $$
>
> for a sufficiently large constant $K>0$, then the EM iterates $\\{\boldsymbol{\phi}^{(t)}\\}_{t=0}^{\infty}$ satisfy the bound
>
> $$
> \||{\boldsymbol{\phi}^{(t)} - \boldsymbol{\phi}^* }\||_2
> \le
> \rho^t{\||\boldsymbol{\phi}^{(0)} - \boldsymbol{\phi}^* }\||_2 + \frac{6K}{1-\rho} \cdot \frac{\max\\{n,m\\} \cdot \ln\left( \frac{4\min\\{n,m\\}}{\delta} \right)}{N}
> $$
>
> with probability at least $1 - \delta$.
>
> We will include this result along with proof in the final version of the paper.
>
> An important direction for future work is to show global convergence: either by showing that there is a single global minima, or by establishing an initialization method that guarantees initialization in the vicinity of the global minimia.

---

> > ### Author Rebuttal · Reviewer_CwQL · 2026-04-03
> >
> > Thanks for the detailed response. My main concerns have been adequately addressed, especially the clarification of the convergence claim and the added implementation details. I will raise my score accordingly, and I have no further questions.

---

> > > ### Author Response · Authors · 2026-04-08
> > >
> > > Thank you very much for reading our rebuttal and adjusting your score! Please let us know if you have any more concerns.

---

### Official Review · Reviewer_ycLk · 2026-03-12

**Soundness:** 3
**Presentation:** 3
**Significance:** 2
**Originality:** 3
**Overall Recommendation:** 4
**Confidence:** 3

**Summary:**

In this paper, the authors investigated the problem of estimating user abilities and item difficulties in the Rasch model, which is a fundamental model in Item Response Theory (IRT). When comparing to the recent study Nguyen & Zhang (2023a), this paper aims to address the efficiency problem by proposing an EM-based algorithm for joint estimation of item and user parameters by introducing PÅLolya-Gamma latent variables, which simplify the logistic log-likelihood. The experiments have been conducted on both Synthetic and real-world datasets.

**Compliance With Llm Reviewing Policy:**

Affirmed.

**Final Justification:**

I mainly update my score to align with other reviewers who has similar or higher confidence score, while I still have concerns about the justification from the authors about the usage of recsys datasets.

**Key Questions For Authors:**

Please find my questions in the above detailed comments, as they could be better understood in related context. Thanks.

**Limitations:**

Yes, some are discussed in Impact statement.

**Strengths And Weaknesses:**

I really like the proposed EM-style algorithm to jointly estimate both user and item parameters, and it is nice that the authors also provide theoretical analysis.

However, I still have some questions/concerns while reading the paper:

The pre-processing of the existing real-world datasets are not clear. For example, there are actual ratings (1-5) in MovieLens datasets, it is unclear how they are converted to 1s and 0s in this study. I am guessing the authors following Nguyen & Zhang (2023a), but this should be made clear.

I also have questions about the assessment of the proposed method’s ability for recommendation tasks. While this proposed method is able to do recommendations like the one formulated in Section 6. However, in the community of recommender systems, more advanced models have been developed for years, e.g. deep-learning based collaborative filtering algorithms and factorisation algorithms. While I understand the paper seems to focus on IRT related techniques, but when claiming its ability for recommendation purposes, I still think it is needed to compare with the state-of-the-art recommendation algorithms in the community.

Moreover, the recommendation tasks used in this paper is limited. For example, after converting the actual ratings to 0s and 1s (I assume the authors follows the work in Nguyen & Zhang (2023a) as this is not detailed in the paper), the recommendation task is actually simplified in this case, as the model only needs to distinguish highly positive and highly negative, which might be the reason why the gaps (in terms of accuracy) among all compared algorithms are relatively small. A consequential question/suggestion is: It would be needed to test the popular and relatively hard recommendation tasks, e.g. actually rating predictions or top-N recommendations (even sequential-recommendations) to really show the performance advantage of the proposed method.

The experiment results seems a bit mixed. For example, “MMLE is the fastest method on BOOK GENOME and ML-20M, but its predictive performance is inferior to PGEM across all metrics”. However, the gaps in terms of predictive performance is very small. Moreover, similar observation can be obtained on small and sparse datasets, for example, SPECTRAL is actually quicker than PGEM on MovieLens100k.

Furthermore, because this paper aims for an efficient algorithm, the question I have is: in actual recommender systems, it is very likely that there are millions of users and items: would this still be too expensive for the proposed method? Is it possible to compare with those recommendation algorithms in terms of efficiency?

Minor comments:
1. Reference should be updated for Nguyen & Zhang (2023a).

---

> ### Author Rebuttal · Authors · 2026-03-31
>
> >The pre-processing of the existing real-world datasets...
>
> We thank the reviewer for this comment. We followed Nguyen & Zhang (2023a) to convert the ratings in MovieLens datasets to 1s and 0s. The rating is considered as 1 if it is greater than the user's mean rating, else it is considered as 0. We will explicitly mention this in the final version.
>
> ---
>
> > The experiment results seems a bit mixed...
>
> While MMLE appears faster in some cases (BOOK-GENOME, ML-20M), this is based on *per-prior* runtime. Since MMLE is known to be sensitive to prior misspecification, we need to tune the prior, for which we follow the same approach as Nguyen & Zhang (2023a): we evaluate 15 priors and select the best that maximizes validation log-likelihood. The times reported in the paper for MMLE are averaged per prior, so the effective runtime scales as:
>
> Total Time $\approx$  #priors $\times$ time per prior,
>
> making it higher than our method in practice. We will clarify this point in the final version to avoid confusion.
>
> Regarding predictive performance, we agree that the margins may appear small but are consistent across all metrics and datasets, and are achieved without prior tuning, making our method simpler and more robust.
>
>
> SPECTRAL is slightly faster than PGEM only on ML-100K (by 0.212s). On larger datasets (BOOK-GENOME, ML-20M, etc.), SPECTRAL is significantly slower: for instance, on BOOK-GENOME, PGEM takes ~7557s vs. 14,312s for SPECTRAL. Further, the ablation study provided in response to Reviewer EgYv shows that if we run our method with $\nu=1.0$, which is a more balanced choice of the regularization parameter, the inference time for PGEM reduces even further.
>
> ---
>
> >It would be needed to test the popular and relatively hard recommendation tasks, e.g. actually rating predictions or top-N recommendations (even sequential-recommendations)..
>
> Following the suggestion, we conduct experiments evaluating recommendation metrics adapted to the IRT setting. For each user, items labeled as 1 in the test set are treated as relevant (ground truth), and all test items, which are unseen during training, are used for evaluation. We evaluate on K = 1 (strictest), K = 4 (moderate) and K = 8 (less strict). We found that our method is competitive even on these metrics.
>
> [Evaluation on recsys metrics](https://anonymous.4open.science/r/ICML26_Rebuttal-6A44/Recsys/Baselines.png)
>
> >.. compare with the SOTA recommendation algorithms.
>
> We would like to clarify that our main goal is to develop an estimation method for latent parameters under the Rasch model in IRT. While IRT has multiple applications, one of which can be in the recsys domain, competing with SOTA in the recsys literature is outside our scope. Recsys literature is very wide with a particular focus on finding rich embeddings/representations for users and items. IRT on the other hand focuses on measuring quality using a single parameter per user and item. Another point of difference is binary responses vs. multi-level ratings.
>
> However, following the suggestion, we compared our method with a collaborative filtering approach: rank-1 Logistic Matrix Factorization (LMF).  In our model, each user/item is associated with a single scalar latent parameter, which effectively gives the binary response matrix a rank-1 structure. Hence, we compare with the corresponding rank-1 method. Since LMF assumes the form $\sigma(\theta\cdot\beta)$ while Rasch model assumes the form $\sigma(\theta-\beta)$, so to ensure a fair comparison, we modify the evaluation metric functions to work with the appropriate logits. The results of this evaluation in the links provided. We find that our method performs better than this baseline.
>
> [PGEM vs. MF](https://anonymous.4open.science/r/ICML26_Rebuttal-6A44/Recsys/PGEM%20vs%20MF.png)
>
> [PGEM vs. MF (continued)](https://anonymous.4open.science/r/ICML26_Rebuttal-6A44/Recsys/PGEM%20vs%20MF%20K%20metrics.png)
>
> While we could technically compare our method against deep learning based approaches, they learn richer embeddings and have higher model capacity, and thus, better expressive power. Hence, we believe it would not be a fair comparison against our method.
>
> Moreover, we use recsys datasets such as MovieLens because they are well-known and standard benchmarks, following prior work (Nguyen & Zhang (2023a)).
>
> >In actual recsys, it is very likely there are millions of users and items... Is it possible to compare with those recommendation algorithms in terms of efficiency?
>
> As mentioned above, our goal is not to compete with the SOTA for recsys. Our method is better suited for crowdsourcing tasks related to recsys, where we evaluate the quality of different raters/items using sparse ratings. For actual recommendation tasks, one would like to leverage deep learning approaches that learn rich representations for users/items. Having said that, our method generally scales better than existing approaches for IRT.

---

> > ### Author Rebuttal · Reviewer_ycLk · 2026-04-02
> >
> > I appreciate the authors' responses. However, my questions about the marginal performance and the baseline (for recsys) still remain. While it is argued that the proposed method is more suitable for crowdsourcing tasks related to recsys, this does not match the usage of those benchmark datasets in the area (e.g. the MovieLens datasets). For the performance, the improvement is really marginal, and on those small datasets, they are mixed indeed.

---

> > > ### Author Response · Authors · 2026-04-05
> > >
> > > We thank the reviewer for the reply.
> > >
> > > > While it is argued that the proposed method is more suitable for crowdsourcing tasks related to recsys, this does not match the usage of those benchmark datasets in the area (e.g. the MovieLens datasets).
> > >
> > > We would like to request the reviewer to judge our work on the basis of its application to the IRT domain in general, rather than specific recommendation tasks. Our response to your comment is in 2 parts: (1) justification for why RecSys datasets were used, (2) additional experiments on non-RecSys datasets.
> > >
> > > **Justification of why RecSys datasets were used**: The main reason we use RecSys datasets is because they have been used in the existing IRT literature (Nguyen & Zhang (2023a)), and deviating from this literature will not be justified. In the IRT setting one expects data with *binary user-task correctness labels*. On the other hand, RecSys datasets have user-item ratings which are more subjective and depend on a rich set of item/user attributes. Nevertheless, we follow Nguyen & Zhang (2023a) in binarizing the RecSys ratings to enable evaluation in the IRT setting. Even though this transformation may lose information, it is not uncommon to see ML datasets designed for a particular task being adapted in the study of a slightly different task.
> > >
> > > **Additional experiments on non-RecSys datasets**: On the reviewer's suggestion, we have also evaluated our method on two other datasets.
> > > 1. ICAR-SAPA ("Condon, David & Revelle, William. (2016). Selected ICAR Data from the SAPA-Project...") is a large-scale **psychometric test** dataset involving participants answering questions on sequences, 3D patterns, etc. and contains total 1,199,896 responses by 95236 participants on 60 items (questions).
> > > 2. MTurk-Lexical dataset ("Ratcliff and Hendrickson (2021). Do data from mechanical Turk subjects replicate...") is a **crowdsourcing** dataset for the lexical decision task administered to MTurk crowdworkers. There are 142 participants and 4588 items (tasks) with total 68160 responses.
> > >
> > > |Dataset|Method|LogLik|AUC|Acc|Brier|Time (s)|
> > > |-|-|-|-|-|-|-|
> > > |ICAR-SAPA|PGEM|**-0.5259**|**0.8137**|**0.7322**|**0.1764**|**117.81**|
> > > ||Spectral|-0.5383|0.8036|0.7243|0.1812|2128.68|
> > > ||JMLE|-0.5979|0.7854|0.7194|0.1899|210.79|
> > > ||MMLE|-0.6044|0.7898|0.7208|0.1919|845.40|
> > > |MTurk-Lexical|PGEM|**-0.2787**|**0.8315**|**0.8912**|**0.0818**|**4.24**|
> > > ||Spectral|-0.2950|0.8111|0.8846|0.0870|4.33|
> > > ||JMLE|-0.3281|0.7997|0.8888|0.0874|33.42|
> > > ||MMLE|-0.3751|0.7887|0.8840|0.0921|60.45|
> > >
> > > Our method again outperforms other methods on all metrics, and is substantially faster on the larger ICAR-SAPA dataset.
> > >
> > > > For the performance, the improvement is really marginal, and on those small datasets, they are mixed indeed.
> > >
> > > We would like to mention two main points: (1) our method is *significantly faster than other methods* especially on large datasets, and (2) shows *consistent* improvement in metrics over almost all datasets. Achieving faster computation time while retaining or improving accuracy metrics is a non-trivial challenge.
> > >
> > > The improvement in metrics in some cases may seem marginal, but these small improvements are quite significant due to the inherent difficulty of this problem and sparsity of the data. On larger datasets such as ICAR-SAPA, MovieLens, BOOK-GENOME, and HETREC, our method clearly outperforms (no ties) all other methods on every metric. On small datasets such as UCI and 3 Grades, all methods may perform similarly: the datasets being very small, the margins are also tight, so it is natural that the results may seem mixed. But the clear pattern even on these is that PGEM is competitive.
> > >
> > > As discussed in the paper, each existing method has significant failure points. JMLE fails when the number of items is small, MMLE fails when the prior is misspecified, CMLE is time-inefficient and inconsistent for large sparse datasets, and Spectral requires dense-regimes evident from the sample complexity proposed in their paper and can be time-inefficient on larger datasets as compared to our proposed method.
> > >
> > > We understand your concern regarding the mixed results on small datasets and will revise the framing to state that our method clearly outperforms baselines on larger datasets, and with significant improvement in computation time.
> > >
> > > > Regarding recsys metrics
> > >
> > > As discussed in the previous response, our goal is not to develop SOTA for RecSys. Experiments show that the joint estimation of parameters in the IRT setup allows the method to recommend items competitively and sometimes better than the baseline [PGEM vs. MF](https://anonymous.4open.science/r/ICML26_Rebuttal-6A44/Recsys/PGEM%20vs%20MF%20K%20metrics.png). Comparison with IRT baselines on recsys metrics also show that our method performs the best (with ties on some metrics) [PGEM vs. IRT baselines](https://anonymous.4open.science/r/ICML26_Rebuttal-6A44/Recsys/Baselines.png).
> > >
> > > ---
> > >
> > > We hope this clarifies your concerns and we welcome further discussion.

---

### Official Review · Reviewer_QiCX · 2026-03-13

**Soundness:** 3
**Presentation:** 2
**Significance:** 2
**Originality:** 1
**Overall Recommendation:** 4
**Confidence:** 3

**Summary:**

This paper presents a novel and computationally efficient approach for the joint estimation of user abilities and item difficulties within the Item Response Theory (IRT) model. The authors effectively address several limitations of existing methods, such as the inconsistent estimation of JMLE, the computational burden of MMLE/CMLE, and the dense data requirements of recent spectral techniques.
Main contributions:
● A Pólya–Gamma augmented EM method for simultaneous user/item estimation.
● A proof of global convergence to the MLE at a linear rate.
● Empirical gains on synthetic and real datasets, notably in sparse-data settings.

**Compliance With Llm Reviewing Policy:**

Affirmed.

**Key Questions For Authors:**

As seen in weakness

**Limitations:**

Authors have illustrated potention impacts of this paper.

**Strengths And Weaknesses:**

strength:
● Methodologically, adapting Polya–Gamma augmentation to derive a clean EM algorithm for joint estimation is elegant and leads to a very simple quadratic M-step (linear system), which is practically attractive and easy to implement.
● Theoretical guarantees are a clear strength.
weakness:
● originality:
The core technical idea—introducing Polya–Gamma latent variables to turn a logistic  likelihood into a quadratic form and then running EM—is conceptually a direct application of Polson et al. (2013). In its current form, the use of Polya–Gamma augmentation inside an EM scheme for logistic latent-variable models is standard, so the originality of the methodological contribution is limited.
Conceptually, The work discusses a broad domain (IRT, recommendation systems, crowdsourcing, etc.), but the actual technical novelty is concentrated on a fairly narrow algorithmic choice. This gap between breadth of application claims and depth of methodological innovation may lead some reviewers to view the contribution as incremental.
● typos:
● L180: must satisy -> must satisfy
● L254: comma at the end of the line
● L374: to make datasets more denser -> to make datasets denser
● Inconsistent Notation:
● Equation (4) on page 2 (and (12) in Appendix B): The factor is should consistently be 2^{-b} or 1 / 2^b, not 1 / (2 b).

---

> ### Author Rebuttal · Authors · 2026-03-31
>
> We thank the reviewer for the valuable feedback. We clarify the concerns as follows:
>
> > Originality of the paper and contributions.
>
> Our paper studies item response theory (IRT) and proposes a new EM algorithm based on the Pólya-Gamma (PG) augmentation method. While the Pólya-Gamma (PG) augmentation technique is well-known, our paper has two main novelties.
>
> Firstly, it was apriori unclear whether PG augmentation can be used to solve the IRT problem, particularly under the Rasch model. Recognizing this applicability is an important contribution of the paper. This connection enables us to develop a mathematically tractable EM algorithm providing closed-form updates in the M-step. This results in a nice simplification over the existing alternate minimization based approaches known in the literature.
>
> Secondly, and more importantly, we provide convergence results for our PGEM algorithm, which to our knowledge is the first theoretical analysis of EM under PG augmentation. These local convergence guarantees are also noted by you as a clear strength. We are also able to obtain finite-sample guarantees (see response to Reviewer CwQL) which are stronger than the known guarantees under the Rasch model. Our analysis and framework for obtaining statistical guarantees may further find application in other contexts.
>
> >The work discusses a broad domain (IRT, recommendation systems, crowdsourcing, etc.), but the actual technical novelty is concentrated on a fairly narrow algorithmic choice.
>
> Our main goal in this paper was to develop an efficient joint learning approach for a general IRT setting under the Rasch Model, which spans various practical applications. Regarding the applicability to a broad domain, we can see improved results over datasets in various domains where our method consistently outperforms other baselines for modeling the Rasch model. This directly translates to the benefits we get with this algorithm in broader domains where IRT is applicable.
>
> ---
>
> We hope that the concerns about novelty, contributions and broad applicability are clarified. We would be happy to engage in further discussion if you have any additional concerns.
>
> ---
>
>
> ### Typos:
> > L180: must satisy -> must satisfy, L254: comma at the end of the line, L374: to make datasets more denser -> to make datasets denser
>
> We thank the reviewer for identifying the typos. We will fix them in the final version.
>
> >Inconsistent Notation: ● Equation (4) on page 2 (and (12) in Appendix B): The factor is should consistently be 2^{-b} or 1 / 2^b, not 1 / (2 b).
>
> We apologize for the confusion. The notation is consistent:
> Equation (4) on page 2 uses 1 / 2^b and (12) in Appendix B uses 2^{-b}. Hence, the notation is consistent.

---

> > ### Author Rebuttal · Reviewer_QiCX · 2026-04-04
> >
> > The rebuttal resolves my concerns about novelty by clearly demonstrating that Pólya-Gamma augmentation is applicable to IRT under the Rasch model, which was previously unclear. This insight enables a tractable EM algorithm with closed-form M-step updates, simplifying prior approaches. Furthermore, the paper provides the first convergence analysis of EM under PG augmentation, addressing my concerns about theoretical grounding. The addition of strong finite-sample guarantees further strengthens the contribution. Overall, my concerns have been addressed.

---

> > > ### Author Response · Authors · 2026-04-08
> > >
> > > Thank for reading our rebuttal! We are happy to hear that our rebuttal was able to resolve all your concerns. In the final version, we will clarify about the contributions and applicability to the broader domain.
> > >
> > > Please let us know if there are any remaining issues or additional concerns we can further clarify.

---

### Official Review · Reviewer_EgYv · 2026-03-14

**Soundness:** 2
**Presentation:** 2
**Significance:** 3
**Originality:** 3
**Overall Recommendation:** 4
**Confidence:** 2

**Summary:**

This paper studies the problem of joint parameter estimation in Item Response Theory (IRT), specifically under the Rasch model. The Rasch model characterizes each user by an ability parameter θ and each item by a difficulty parameter β, where the probability of a correct response follows a logistic function: P(X=1|θ,β) = 1/(1+exp(-(θ-β))).

Previous work includes: joint maximum likelihood estimation (JMLE) which lacks theoretical guarantees and performs poorly on small datasets; marginal and conditional maximum likelihood methods (MMLE, CMLE) which are either prone to prior misspecification or computationally expensive; and spectral approaches that require dense observation regimes.

The authors propose an EM-based algorithm for joint estimation of both user and item parameters by introducing Pólya-Gamma latent variables, which simplify the logistic log-likelihood. Theoretical analysis demonstrates that the EM method converges to the optimum at a linear rate. This approach addresses the computational inefficiency of existing methods while maintaining theoretical soundness, and is designed to perform well in sparser regimes where fewer user-item labelings are available.

The paper validates the proposed method on both synthetic and real-world datasets, demonstrating consistent improvements over existing baselines in terms of estimation accuracy and computational efficiency. Experimental analysis includes sufficient metrics and reasonable baseline selections.

**Compliance With Llm Reviewing Policy:**

Affirmed.

**Final Justification:**

The rebuttal addressed my questions.

**Key Questions For Authors:**

What is the generality of the theoretical assumptions?

What are the key parameters of the PGEM algorithm, and how do they affect the performance?

**Limitations:**

The paper could be improved in the following aspects:

1. More reasonable justification of theoretical assumptions: The authors should provide clearer explanations for the theoretical assumptions and their applicability.

2. Parameter sensitivity analysis: The authors should include ablation studies on PGEM algorithm parameters to analyze their impact on performance.

**Strengths And Weaknesses:**

Soundness

Strengths:

This paper studies the problem of efficient estimation of user and item parameters in item response theory, addressing the high computational complexity of existing methods.

The authors propose an EM algorithm based on the Pólya-Gamma augmentation technique.

Theoretical analysis demonstrates that the EM method converges to the optimum at a linear rate.

Experiments on both synthetic and real-world datasets validate that the method outperforms previous approaches in terms of both efficiency and effectiveness.· The experimental analysis includes sufficient metrics and reasonable baseline selections.

Weaknesses:

The paper does not explain or justify the generality of the theoretical assumptions.

The paper does not provide key parameters of the PGEM algorithm or ablation study results on different parameters.

Presentation

Strengths:

The logic of the paper is relatively clear.

Weaknesses:

The paper contains too many mathematical symbols and notations.

Significance

In the field of item response theory, this paper proposes a novel EM algorithm with theoretical guarantees.

The experimental results validate the effectiveness of the proposed method.

Originality

This paper studies item response theory and proposes a new EM algorithm based on the Pólya-Gamma augmentation method.

This work represents an innovative application of existing methods to the item response theory field.

---

> ### Author Rebuttal · Authors · 2026-03-31
>
> We thank the reviewer for their review. We clarify the main concerns as follows:
>
> > What is the generality of the theoretical assumptions?
> More reasonable justification of theoretical assumptions: The authors should provide clearer explanations for the theoretical assumptions and their applicability.
>
> Firstly, note that the theoretical assumptions are only required to prove the convergence result but the algorithm works even if these are not satisfied. Having said that, these theoretical assumptions are quite mild and easy to satisfy. The main assumption is $\nu\ge0.5$ in Theorem 5.1 [Lines 202-215], which is enough to show that properties 1-4 (Balakrishnan et al.) are satisfied under our setting. These properties lead to contraction of the M-operator which enables us to prove the local population-level convergence guarantees. The convergence rate is governed by the contraction coefficient $\rho=\frac{2}{5 \nu(4\nu-1)}$. As $\nu$ increases, the denominator $5\nu(4\nu-1)$ grows quadratically, leading to a monotonic decrease in $\rho$, enabling faster convergence.
>
> As detailed in the following ablation study, the above theoretical analysis also has implications for practice. Our PGEM converges faster when $\nu=1.0$ than when $\nu=0.25$ with some improvement in metrics as well.
>
> ---
>
> > What are the key parameters of the PGEM algorithm, and how do they affect the performance?
> Parameter sensitivity analysis: The authors should include ablation studies on PGEM algorithm parameters to analyze their impact on performance.
>
> The key parameter of the PGEM algorithm is the (a) regularization parameter $\nu$. Other parameters include:
>
> (b) $EM_\text{max-iter}$: Maximum number of iterations the algorithm runs for. We conservatively set it to $100$, though in practice convergence typically occurs much earlier when $\nu$ is appropriately set ($\nu\ge0.5$). The ablation study shows convergence within 51 iterations for $\nu=1.0$ across all datasets.
>
> (c) $GS_\text{max-iter}$: Number of Gauss-Seidel iterations used to solve the M-step linear system. Empirically, we observed that across all datasets, a single iteration suffices for good performance which can also be seen in the Ablation Study. Thus, in our experiments, we have set this parameter to $1$.
>
> (d) $EM_\text{loglik-tolerance}$: Log-likelihood convergence tolerance, set to $1\mathrm{e}{-5}$.
>
> (e)  $GS_\text{var-tolerance}$: M-step convergence tolerance for $\mathbf{\beta}$ and $\mathbf{r}$, set to $1\mathrm{e}{-3}$.
>
> The ablation study shows the importance of these parameters. It suggests that our method is stable under reasonable parameter variations, with $\nu$ playing a significant role in controlling the convergence rate.
>
> ---
>
> ## Ablation Study
>
> **Dataset: ML-1M**
>
> [Ablation for Regularization parameter $\nu$](https://anonymous.4open.science/r/ICML26_Rebuttal-6A44/Ablations/Regularization.png)
> |$\nu$|loglik|acc|auc|brier|time (sec)|
> |-|-|-|-|-|--|
> |0.01|-0.580206|0.696367|0.759402|0.197726|35.019382|
> |0.05|-0.579784|0.696441|0.759448|0.197683|34.585406|
> |0.10|-0.579560|0.696441|0.759498|0.197643|33.614114|
> |0.25|-0.579191|0.696446|0.759609|0.197558|30.793731|
> |0.50|-0.578862|0.696431|0.759721|0.197471|22.969082|
> |1.00|**-0.578599**|**0.696495**|**0.759796**|**0.197406**|19.372000|
> |1.50|-0.578601|0.696490|0.759758|0.197424|**18.077930**|
>
> We can see that increasing $\nu$ leads to faster convergence. Subsequently, we rerun the experiments with $\nu=1.0$ which is a balanced choice of the regularization parameter, and we report the results in the Updated Results link.
>
> [](https://anonymous.4open.science/r/ICML26_Rebuttal-6A44/Ablations/Dataset%20vs%20Iters.png)
>
> |Dataset|Iters ($\nu$=1.0)|Iters ($\nu$=0.25)|
> |-|-|-|
> |HETREC|38|76|
> |BOOK GENOME|49|NA|
> |ML-100K|18|26|
> |ML-1M|29|45|
> |ML-10M|45|NA|
> |ML-20M|51|NA|
>
> Hence, setting $\nu=1.0$ for others, we can observe that at most 51 iterations are needed for convergence for these datasets.
> For $\nu=0.25$, the results show that for BOOK-GENOME, ML-10M and ML-20M convergence does not happen within 100 iterations. These results align with the convergence guarantee which requires $\nu\ge0.5$ (Theorem 5.1, Lines 202-215).
>
> [Ablation for GS max iter](https://anonymous.4open.science/r/ICML26_Rebuttal-6A44/Ablations/GS_max_iter.png)
>
> [Ablation for Tolerance parameters](https://anonymous.4open.science/r/ICML26_Rebuttal-6A44/Ablations/Tolerances.png)
>
> Metrics change negligibly varying GS max iter and tolerance parameters.
>
> ---
>
> Based on the ablation study, we rerun the experiments with $\nu=1.0$, keeping other parameters unchanged.
>
> [Updated Results](https://anonymous.4open.science/r/ICML26_Rebuttal-6A44/Ablations/Updated_Results.png)
>
> With these updates:
> 1. On ML-100k, PGEM now outperforms competing methods in terms of runtime.
> 2. On ML-10M, PGEM slightly outperforms MMLE in Accuracy (0.6936 vs. 0.6930) and Brier Score (0.1994 vs. 0.2003).

---

> > ### Author Rebuttal · Reviewer_EgYv · 2026-04-01
> >
> > Thanks for the rebuttal, which addressed my questions.

---

> > > ### Author Response · Authors · 2026-04-08
> > >
> > > Thank for reading our rebuttal! We are happy to hear that our rebuttal was able to resolve all your concerns. We will be including ablation studies and clarify the generality of theoretical assumptions in the final version.
> > >
> > > Please let us know if there are any remaining issues or additional concerns we can further clarify.

---

### Decision · Program_Chairs · 2026-04-30

**Decision:**

Accept (regular)

**Comment:**

This paper studies joint parameter estimation in Item Response Theory (IRT), focusing on the Rasch model. The authors propose an EM-based algorithm that introduces Pólya–Gamma latent variables to facilitate the joint estimation of user and item parameters by simplifying the logistic log-likelihood. The theoretical analysis shows that the proposed EM algorithm converges to the optimum at a linear rate. Experimental results on both synthetic and real-world datasets demonstrate that the method outperforms existing approaches in terms of efficiency and effectiveness. The empirical evaluation includes appropriate metrics and reasonable baseline comparisons. Most of the reviewers' concerns have been addressed in the rebuttal. Nevertheless, the authors are encouraged to incorporate the reviewers' suggestions (e.g., reporting confidence intervals) in the final version.